# SRPO: Enhancing Multimodal LLM Reasoning via Reflection-Aware Reinforcement Learning

**Zhongwei Wan**[2†∗]   **Zhihao Dou**[3†]   **Che Liu**[4]   **Yu Zhang**[11]   **Dongfei Cui**[5]
**Qinjian Zhao**[6]   **Hui Shen**[7]   **Jing Xiong**[10]   **Yi Xin**[12]   **Yifan Jiang**[8]
**Chaofan Tao**[10]   **Yangfan He**[9]   **Mi Zhang**[2]   **Shen Yan**[1]

[1]ByteDance Seed    [2]The Ohio State University
[3]Case Western Reserve University    [4]Imperial College London    [5]Duke University
[6]Kean University    [7]University of Michigan [8]University of Southern California    [9]University of Minnesota    [10]The University of Hong Kong    [11]Tongji University    [12]Nanjing University
Correspondence to: wan.512@osu.edu, sheny@bytedance.com
https://srpo.pages.dev

## Abstract

Multimodal large language models (MLLMs) have shown promising capabilities in reasoning tasks, yet still struggle significantly with complex problems requiring explicit self-reflection and self-correction, especially compared to their unimodal text-based counterparts. Existing reflection methods are simplistic and struggle to generate meaningful, instructive feedback, as the reasoning ability and knowledge limits of pre-trained models are largely fixed during initial training. To overcome these challenges, we propose *multimodal **S**elf-**R**eflection enhanced reasoning with Group Relative **P**olicy **O**ptimization* **SRPO**, a two-stage reflection-aware reinforcement learning (RL) framework explicitly designed to enhance multimodal LLM reasoning. In the first stage, we construct a high-quality, reflection-focused dataset under the guidance of an advanced MLLM, which generates reflections based on initial responses to help the policy model to learn both reasoning and self-reflection. In the second stage, we introduce a novel reward mechanism within the GRPO framework that encourages concise and cognitively meaningful reflection while avoiding redundancy. Extensive experiments across multiple multimodal reasoning benchmarks—including MathVista, MathVision, Mathverse, and MMMU-Pro—using Qwen-2.5-VL-7B and Qwen-2.5-VL-32B demonstrate that SRPO significantly outperforms state-of-the-art models, achieving notable improvements in both reasoning accuracy and reflection quality.

## 1   Introduction

Multimodal reasoning is central to numerous real-world scenarios, such as interpreting scientific figures, geometric reasoning, and integrated complex image-text comprehension tasks [1–3]. Although recent approaches have attempted to transfer effective RL-based reasoning methods [4–7] from textual models to multimodal scenarios [8–11], these methods generally encounter considerable limitations. Specifically, existing MLLMs typically follow a token-level Markov process [12, 13] during generation, which relies on local dependencies. This often leads to redundant, repetitive, or erroneous reasoning steps in their output [14]. Such issues hinder reasoning models from achieving significant improvements over fast-thinking models; in some cases, their performance is even inferior. For instance, GPT-o1, despite its explicitly structured reasoning pathways, achieves slightly lower

---

∗Project Lead (work done during internship at ByteDance). †Equal Contribution.

39th Conference on Neural Information Processing Systems (NeurIPS 2025).

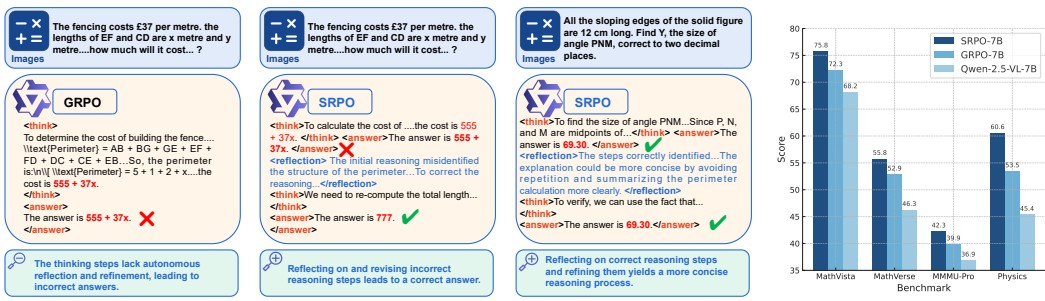

Figure 1: Left: Illustrative examples of reflection improving reasoning. Right: Quantitative comparison on benchmark datasets.

accuracy (73.9%) on MathVista compared to Qwen2.5-VL-72B (74.8%) [1, 15]. The primary reason lies in the presence of incorrect and redundant steps, which negatively affect final performance.

Previous studies have shown that self-reflection [16, 17] is an effective approach to address this issue. By explicitly encouraging the model to review, evaluate, and revise its own reasoning process, self-reflection helps eliminate unnecessary or incorrect steps, enhances logical coherence, and promotes deeper understanding [18]. However, recent empirical studies [18–20] indicate that the upper bounds of reasoning capabilities in pre-trained models are largely established during the initial pre-training phase. Consequently, these studies indicate that reinforcement learning improves reasoning by activating decision-making within fixed structures, rather than enabling the acquisition of new knowledge or behaviors. To effectively surpass these inherent limitations, external interventions such as advanced reflective experiences or cognitively guided techniques are required. While previous approaches [18, 16, 20] have attempted to enhance self-reflective reasoning through direct prompting or reinforcement learning, their effectiveness remains limited by the constraints imposed during pre-training, making them insufficient for substantially improving reflective reasoning and overall reasoning performance.

Therefore, *designing effective enhancement strategies to improve the intrinsic reasoning capabilities of MLLMs beyond the constraints established during pre-training remains a challenging problem.* To address this question, inspired by cognitive science emphasizes that robust human reasoning involves active self-reflection and iterative self-correction steps [21–23, 16], we integrate explicit reflective methods within ***both multimodal Supervised Fine-Tuning (SFT) and RL***, enabling models to surpass their intrinsic reasoning boundaries established in the pre-training phase. Unlike previous studies [24–26], which focus solely on enhancing reasoning ability by aligning with extended chain-of-thought supervision, our goals are not only strengthens the model's reasoning performance but also fosters its capacity for self-reflection.

Motivated by these insights, we introduce **SRPO** (*multimodal **S**elf-**R**eflection enhanced reasoning with Group Relative **P**olicy **O**ptimization*), a novel two-stage reflective training framework specifically designed to promote explicit self-reflection and self-correction behaviors within MLLMs. **(i) In the first stage**, we utilize an advanced MLLM to generate reflection content based on the discrepancies between the policy model's outputs and the ground truth. In this process, the model autonomously evaluates its multiple generated responses, identifies errors, and iteratively revises them through reflective reasoning. Subsequently, we leverage these high-quality reflection datasets to perform multimodal reflection-based supervised fine-tuning (SFT), providing a cold-start initialization for subsequent reinforcement learning. **(ii) In the second stage**, we further propose a reflection-aware RL method built upon the Group Relative Policy Optimization (GRPO) algorithm [4]. Our specifically designed reward function actively incentivizes concise, task-oriented reflection steps, explicitly punishing overly verbose or redundant reflections, thus effectively encouraging MLLMs to adopt meaningful reflective behaviors via RL stage. As illustrated in Figure 1, after two-stage training, SRPO enables MLLMs to autonomously generate reflective reasoning, effectively refine intermediate thinking steps, and consequently achieve improved reasoning performance across various benchmarks compared to the GRPO.

We conduct comprehensive experiments across several widely adopted multimodal reasoning benchmarks, including MathVista [1], MathVison [27], and MMMU-Pro [3], utilizing representative multimodal models (e.g., Qwen-2.5-VL-7B and Qwen-2.5-VL-32B [28]). Results demonstrate that SRPO consistently and significantly outperforms current state-of-the-art models, achieving notable

improvements in reasoning accuracy, reflection quality, and cross-task generalization. These empirical findings provide strong evidence that explicit reflection-oriented training can effectively extend multimodal models' reasoning capabilities beyond the inherent cognitive boundaries set during pre-training. Our core contributions are summarized as follows:

- **Novel reflection-oriented SFT construction.** We introduce a novel reflective data generation pipeline that leverages the original model's responses. By using a large MLLM (e.g., GPT-o4-mini [7]), we generate corresponding reflection processes aligned with the gold-standard answers. This pipeline is designed to teach the policy model both effective reasoning and reflective thinking.

- **Reflection-aware reinforcement learning.** We develop a tailored GRPO-based RL method (**SRPO**) equipped with an explicit reward function to incentivize meaningful reflective reasoning.

- **Empirical validation and insights.** Extensive evaluations across various multimodal reasoning benchmarks confirm that SRPO achieves state-of-the-art performance, clearly demonstrating the effectiveness of self-reflection enhancements in multimodal reasoning contexts.

## 2 Related Works

**Reinforcement Learning for LLM Reasoning.** Recent advancements in large-scale RL, such as DeepSeek-R1 [4], have demonstrated substantial progress in enhancing complex, human-like Chain-of-Thought (CoT) reasoning by utilizing result-oriented or formatting-specific reward signals. In parallel, several studies, including Open-Reasoner-Zero [29], SimpleRL-Zoo [30], AlphaMed [31], and Logic-RL [32], have explored directly fine-tuning base language models using RL without any supplementary supervised fine-tuning stages. Additionally, methods such as Light-R1 [33] and DeepScaler [34] introduce specially constructed cold-start datasets designed explicitly to encourage detailed step-wise reasoning during initial training phases. Meanwhile, recent analyses [18, 20, 19] have also shed light on intrinsic limitations of purely RL-based reasoning enhancement strategies. Furthermore, complementary approaches such as VAPO [35], DAPO [12], and Dr. GRPO [36] have sought to refine the Group Relative Policy Optimization (GRPO) framework by optimizing reward design and enhancing advantage estimation techniques, thus more effectively promoting deeper reasoning behaviors within language models. In contrast, our work specifically targets multimodal complex reasoning, explicitly emphasizing self-reflection or correction to enhance reasoning performance during both multimodal SFT and RL training phases.

**Reinforcement Learning for Multimodal LLM Reasoning.** State-of-the-art multimodal reasoning capabilities are largely dominated by proprietary models, such as GPT-o3 and o4 [7], Gemini-2.5-Pro-T [5], and Seed1.5-VL-T [37]. Recent studies aim to close this gap via reinforcement learning (RL) on open-source multimodal LLMs. LMM-R1 [9] introduces a two-stage, rule-based RL, though mainly benefiting textual scenarios. Reason-RFT [38] leverages supervised fine-tuning (SFT) with Chain-of-Thought (CoT) data to initialize RL. Vision-R1 [26] enhances multimodal CoT datasets using DeepSeek-R1 and employs progressive thinking suppression in GRPO training. MM-Eureka [8] presents the MMK12 dataset alongside a two-stage RL method, while VL-Rethinker [15] utilizes selective sample replay and explicit textual rethinking triggers to refine multimodal reasoning. R1-V [39] explores RL primarily within visual-centric reasoning tasks but has limited generalization to broader multimodal domains. However, none of these approaches explicitly emphasize self-reflection or correction during both SFT and RL training phases, resulting in suboptimal reasoning performance. Furthermore, poorly designed reward functions leave these methods vulnerable to length redundancy.

## 3 Method of SRPO

In the following sections, we present the detailed methodology of our **SRPO** training framework, emphasizing our two core contributions: **(1) Novel reflection-oriented SFT data construction.** In this stage, we construct a reflection dataset to inject reflective capabilities into the policy model. Through training on this dataset, we aimed to achieve two goals: first, to enhance the policy model's ability for self-reflection and self-correction during cold-start initialization; and second, to effectively transfer the reflective knowledge of large-scale MLLMs into the policy model, enabling it to learn how to reflect effectively gradually. **(2) Reflection-aware reinforcement learning**, where we propose a tailored GRPO-based RL algorithm, **SRPO**, equipped with a reflection-aware reward function that promotes reflective reasoning.

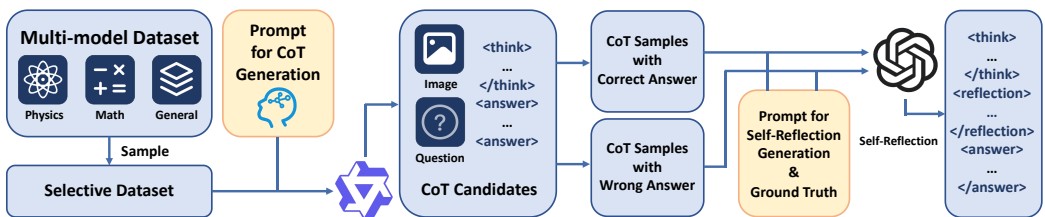

Figure 2: Pipeline of Self-Reflection SFT data construction, including CoT and self-reflection generation.

## 3.1 Reflection-oriented Cold-start Initialization

### 3.1.1 Self-Reflection SFT data construction

**Motivation.** To address the limitations of local dependency in MLLM reasoning, often resulting in redundant, incoherent, or incorrect outputs, self-reflection [16] becomes essential for improving reasoning quality. However, the absence of reflection knowledge and skills can lead to low-quality or superficial self-correction. RL methods typically guide a model towards selecting high-reward reasoning paths already represented within its intrinsic knowledge distribution, rather than inducing genuinely novel cognitive capabilities or knowledge [19]. In contrast, by incorporating external knowledge distillation, supervised fine-tuning (SFT) has been shown to effectively expand the cognitive boundaries of reasoning [20]. To this end, we propose a supervised fine-tuning (SFT) approach that explicitly injects reflection knowledge into the policy model. By learning from high-quality reflective examples, the model acquires the ability to identify, diagnose, and revise flawed reasoning, ultimately enhancing its coherence, efficiency, and self-awareness during the reasoning.

**Less is More.** To support effective reflection learning, we construct a high-quality dataset generated by advanced MLLMs, containing two types of reflective examples: one for refining correct CoTs by removing redundancy, and another for revising incorrect CoTs through error correction. While prior SFT approaches [24–26] focus on mimicking correct reasoning, our reflection-oriented SFT explicitly injects reflection knowledge, enabling the model to detect flaws and refine its reasoning. These approaches are complementary—ours enhances reasoning correction and self-improvement. As illustrated in Figure 2, we begin by curating a high-quality subset of $N = 10,000$ multimodal reasoning samples from three large-scale datasets: LLaVA-CoT [25] (100K), Mulberry [24] (260K), and MathV360K [40]. These samples cover diverse domains including physics, mathematics, and general knowledge. Based on this subset, we construct our self-reflection dataset through two complementary strategies: **(1) Refinement of correct CoTs**, and **(2) Revision of incorrect CoTs**. For each sample, we first obtain the initial response generated by the policy model through CoT prompting. Then, using the ground truth answer as guidance, we employ a larger MLLM (e.g., GPT-o4-mini) to generate a self-reflection that either revises flawed reasoning or streamlines correct but verbose outputs. Each final sample thus contains three components: the initial response, the generated self-reflection, and the ground truth answer. In our curated data, approximately 30% of initial responses are correct, while the remaining 70% contain reasoning errors, highlighting the necessity of self-reflection for both wrong solution correction and right question refinement.

### 3.1.2 Cold-start Initialization of Policy Model

This phase equips the policy model $\pi_\theta$ (initial MLLMs) with fundamental self-reflective reasoning capabilities, ensuring it can generate proper reflection-aware reasoning paths before reinforcement learning:

$$\mathcal{L}_{\text{cold-start}} = -\mathbb{E}_{\tau \sim \mathcal{D}} \left[ \sum_{t=1}^{T} \log \left( \pi_{initial}(a_1, <\texttt{reflection}> ... </\texttt{reflection}>, a_2 \mid q) \right) \right]. \tag{1}$$

Here, $a_1$ is the policy model's initial response, $\langle\texttt{reflection}\rangle...\langle/\texttt{reflection}\rangle$ denotes the reflection generated by a large LLM, and $a_2$ is the ground truth answer. Given the input prompt $q$ and the policy model $\pi_{\text{initial}}$, the objective is twofold: (1) to train the model to revise $a_1$ toward $a_2$ using the reflection $\langle\texttt{reflection}\rangle$, and (2) to leverage the reasoning and knowledge embedded in $a_2$ to guide future predictions. This reflection-driven learning process equips the policy model with self-correction capabilities and improves alignment with correct reasoning trajectories.

## 3.2 Reflection-aware Reinforcement Learning

Under the reinforcement learning framework, complex reasoning tasks often leverage Chain of Thought (CoT) steps to improve prediction accuracy and interpretability [4, 6]. However, simply encouraging CoT generation can result in redundant or misleading reasoning [18–20]. To address this, recent work introduces self-reflection, allowing models to revise their reasoning after generation and improve overall quality. Yet without proper control, models may exploit reflection—e.g., by inflating length or complexity for higher rewards without real gains. To counter this, we propose **SRPO**, a framework that enhances both reasoning and reflection through carefully designed reward signals that discourage superficial behaviors.

### 3.2.1 Group Relative Policy Optimization (GRPO)

We utilize Group Relative Policy Optimization (GRPO), following recent advances [4], to optimize our RL-based training. Unlike SFT, which uses token-level losses, GRPO leverages policy gradients calculated from reward losses for effective policy optimization. GRPO promotes exploration of richer and more diverse reasoning solutions by comparing generated responses within sampled groups. Formally, let $Q$ be the question set, $\pi_{\theta_{old}}$ be the policy model, and $\{o_1, o_2, \ldots, o_G\}$ be a group of responses from $\pi_{\theta_{old}}$ for question $q$. Let $\pi_{\theta_{ref}}$ denote the frozen reference model. The GRPO optimization objective is defined as follows:

$$J_{\text{GRPO}}(\theta) = \mathbb{E}_{q \sim Q, \{o_i\}_{i=1}^G \sim \pi_{\theta_{\text{old}}}}$$

$$\left[ \frac{1}{G} \sum_{i=1}^{G} \sum_{t=1}^{|o_i|} \min\left( \frac{\pi_\theta(o_{i,t}|q)}{\pi_{\theta_{\text{old}}}(o_{i,t}|q)} A_i, \text{clip}\left( \frac{\pi_\theta(o_{i,t}|q)}{\pi_{\theta_{\text{old}}}(o_{i,t}|q)}, 1 - \epsilon, 1 + \epsilon \right) A_i \right) - \beta D_{\text{KL}}(\pi_\theta \| \pi_{\text{ref}}) \right] \quad (2)$$

Here, $\epsilon$ and $\beta$ are clipping hyper-parameters and KL-divergence penalty coefficients, respectively. The advantage $A_i$ for each response is computed as:

$$A_i = \frac{r_i - \text{mean}(\{r_1, r_2, \ldots, r_G\})}{\text{std}(\{r_1, r_2, \ldots, r_G\})}, \quad \text{where} \quad \{r_i\}_{i=1}^G \quad \text{are rewards from the group.} \quad (3)$$

GRPO thus replaces the critic model traditionally required in PPO with a computationally efficient intra-group advantage estimation.

### 3.2.2 Enhanced GRPO with Reflection-Aware Rewards (SRPO)

In SRPO, we aim to achieve two main goals: (1) enhance the policy model's reasoning ability through RL, and (2) strengthen its capacity for self-reflection. To realize these goals, we introduce an enhanced reward function that specifically targets the reflection process within the CoT reasoning framework. The format of this response can be summarized as **first solution → reflection → second refined solution**. The total reward $R_{\text{total}}$ can be shown as:

$$R_{\text{total}} = R_{\text{task}} + R_{\text{reflection}}. \quad (4)$$

**Task Reward.** The task-specific reward $R_{\text{task}}$ combines a format reward and an accuracy reward. The format reward $R_{\text{format}}$ encourages the model to enclose its reasoning process within `<think>...</think>` tags. The accuracy reward $R_{\text{accuracy}}$ verifies whether the predicted answer matches the ground truth, guiding the model to produce logically consistent and correct outputs for the first solution. This repeated supervision helps enhance the model's reasoning ability. Formally, $R_{\text{accuracy}}$ is defined as:

$$R_{\text{task}} = R_{\text{format}} = \begin{cases} 0.5, & \text{if format is correct} \\ 0, & \text{otherwise} \end{cases} + R_{\text{accuracy}} = \begin{cases} 0.5, & \text{if first solution matches gt label} \\ 0, & \text{otherwise} \end{cases}$$

$$(5)$$

**Reflection Reward.** Let $I_{\text{ref}} \in \{0, 0.25\}$ indicates proper formatting of reflection segments (enclosed with `<reflect>` tags) , $L_{\text{response}}$ is the total response token length, $T_{\text{target}}$ is the optimal length for all responses and $T_{\text{max}}$ is maximum response lengths respectively. It is noted that the response contains the first solution, reflection and the second solution guided by reflection. The reward can be defined as:

$$R_{\text{reflection}} = I_{\text{eff}} + I_{\text{ref}} + \alpha \, f_{\text{len}}(L_{\text{response}}), \quad (6)$$

where the reflective brevity reward $f_{\text{len}}(L_{\text{response}})$ is explicitly defined to encourage appropriate lengths which can achieve exact and brief results:

$$f_{\text{len}}(L_{response}) = \left(\exp\left(-\frac{|L - T_{\text{target}}|}{T_{\text{max}} - T_{\text{target}}}\right)\right)^2.$$

(7)

The reward $f_{\text{len}}$ peaks at a target length, encouraging concise, informative reasoning, and decays smoothly toward zero as length approaches a defined maximum. This softly constrains output within a desirable range without hard cutoffs. We adopt this exponential form for its simplicity, differentiability, and stable gradient behavior during training.

Additionally, the effectiveness indicator $I_{\text{eff}}$ provides extra rewards if the reflection genuinely improves reasoning outcomes, measured by improvement in the correctness or accuracy of final answers post-reflection:

$$I_{\text{eff}} = \begin{cases} 0.25, & \text{if reflection keeps a corrected answer,} \\ 0.5, & \text{if reflection corrects the wrong answer,} \\ 0, & \text{if reflection fails to correct the wrong answer,} \\ -0.25 & \text{if reflection misconducts the right into wrong answer.} \end{cases}$$

(8)

The proposed reward function $I_{\text{eff}}$ focuses on the second solution results and assesses the reflection's impact on answer correctness in four cases: preserving a correct answer yields +0.25, successfully correcting an incorrect answer results in +0.5, failing to fix a wrong answer receives no reward, and misleading a correct answer incurs -0.25. This design encourages the model to treat reflection not as a formality, but as a tool for improving reasoning quality and avoiding redundancy.

**Advantages over Standard GRPO.** Compared with standard GRPO—which primarily relies on sparse task-level accuracy supervision—our enhanced reflection-aware SRPO framework introduces several critical improvements: **(i)** By enforcing structured reflection formatting through the $I_{\text{ref}}$ indicator, the model is guided to produce consistently well-organized and identifiable reflection segments. **(ii)** The introduction of a smooth, differentiable length reward $f_{\text{len}}(L_{\text{ref}})$ encourages the generation of reflections that are concise yet informative, avoiding hard cutoffs while softly constraining verbosity. **(iii)** The effectiveness reward $I_{\text{eff}}$ directly aligns reward signals with functional improvement, providing positive incentives only when the reflection corrects errors or preserves correctness, and penalizing harmful reflections. **(iv)** By explicitly rewarding reflection utility rather than mere presence, our approach discourages reward gaming behaviors such as empty or verbose reflections, leading to more meaningful reasoning supervision. Together, these enhancements enable SRPO to foster deeper self-correction capabilities, improve sample efficiency, and achieve superior performance in complex reasoning tasks compared to standard GRPO.

## 4 Experiment

### 4.1 Experiment Settings

**Training Dataset. (1) SFT:** To construct the self-reflection SFT dataset for the cold-start initialization phase, we first curate samples from several established multimodal reasoning sources, including the Mulberry dataset (260K) [24], MathV360K [40], and LLaVA-CoT dataset (100K) [25]. We then apply the data construction procedure detailed in Section 3.1.1, ultimately resulting in a refined SFT dataset comprising approximately 10K samples. **(2) RL:** For the subsequent reinforcement learning phase, we aggregate a diverse collection of multimodal reasoning samples from multiple datasets, such as ScienceQA [41], Geometric Math QA [42], ChartQA [43], DVQA [44], AI2D [45], MATH [46], Virgo [47], R1-OneVision [11], MMK12 [8], and PhyX [48]. These datasets collectively encompass mathematical reasoning, general scientific reasoning, and general chart comprehension tasks. The RL training dataset consists of diverse, cross-domain reasoning samples. More details about SFT and RL training dataset collection are shown in Appendix B.2.

**Baselines and Benchmarks.** To comprehensively evaluate SRPO, we compare against three groups of baselines: (1) **Closed-source MLLMs**: General-purpose models GPT-4o [49], Claude3.7-Sonnet [50], Gemini2-flash [51], and the reasoning-optimized GPT-o1 [52]; (2) **Open-source general MLLMs**: Instruction-tuned multimodal models InternVL2.5 [53] and Qwen-2.5-VL [28], ranging from 7B to 78B parameters; and (3) **Open-source reasoning MLLMs**: Explicitly fine-tuned reasoning models,

Table 1: Comparison between our 7B and 32B models, closed-source baselines, and other vision–language models. [†] reproduced by us. The results of other baselines are obtained from their official reports. Bold indicates the best-performing open-source model.

| Model | Math-Benchmark | | | | | General-Benchmark | | |
|---|---|---|---|---|---|---|---|---|
| | MathVista | MathVerse | MathVision | OlympiadBench | WeMath | MMMU-Pro | MMMU | EMMA |
| **Closed-Source MLLMs** | | | | | | | | |
| Claude3.7-Sonnet | 66.8 | 52.0 | 41.3 | 48.9 | 72.6 | 51.5 | 68.3 | 35.1 |
| GPT-4o | 63.8 | 50.2 | 30.4 | 35.0 | 68.8 | 51.9 | 69.1 | 32.7 |
| GPT-o1 | 73.9 | 57.0 | 60.3 | 68.0 | 98.7 | 62.4 | 78.2 | 45.7 |
| Gemini2-flash | 70.4 | 59.3 | 41.3 | 51.0 | 71.4 | 51.7 | 70.7 | 33.6 |
| Seed1.5-VL-T | 85.6 | - | 68.7 | 65.0 | - | 67.6 | 77.9 | - |
| **Open-Source General MLLMs (7B-16B)** | | | | | | | | |
| InternVL2-8B | 58.3 | 22.8 | 17.4 | [†]10.1 | [†]47.2 | 29.0 | 51.2 | 19.8 |
| InternVL2.5-8B | 64.4 | 39.5 | 19.7 | 12.3 | 53.5 | 34.3 | 56.0 | [†]20.6 |
| QwenVL2-7B | 58.2 | 19.7 | 16.3 | [†]9.7 | [†]51.6 | 30.5 | 54.1 | 20.2 |
| Llava-OV-7B | 63.2 | 26.2 | [†]18.5 | [†]8.5 | [†]49.9 | 24.1 | 48.8 | 18.3 |
| Kimi-VL-16B | 68.7 | 44.9 | 21.4 | – | – | – | 55.7 | – |
| QwenVL2.5-7B | 68.2 | 46.3 | 25.1 | 20.2 | 62.1 | 36.9 | 54.3 | 21.5 |
| **Open-Source Reasoning MLLMs (7B)** | | | | | | | | |
| MM-Eureka-8B[1] | 67.1 | 40.4 | 22.2 | 8.6 | [†]55.7 | 27.8 | 49.2 | [†]21.5 |
| R1-VL-7B | 63.5 | 40.0 | 24.7 | [†]10.8 | [†]53.8 | 7.8 | 44.5 | 8.3 |
| R1-Onevision-7B | 64.1 | 46.4 | 23.5 | 17.3 | 61.8 | 21.6 | – | 20.8 |
| OpenVLThinker-7B | 70.2 | 47.9 | 25.3 | 20.1 | 64.3 | 37.3 | 52.5 | 26.6 |
| VL-Rethinker-7B | 74.9 | 54.2 | 32.3 | [†]20.5 | [†]70.2 | 41.7 | 56.7 | **29.7** |
| Vision-R1-7B | 73.5 | 52.4 | [†]27.2 | [†]19.4 | [†]62.9 | [†]37.7 | [†]54.7 | [†]22.4 |
| MM-Eureka-7B[2] | 73.0 | 50.3 | 26.9 | 20.1 | 66.1 | [†]37.6 | [†]55.2 | [†]23.5 |
| ★ (Ours - **SRPO-7B**) | **75.8** | **55.8** | **32.9** | **22.8** | **71.6** | **42.3** | **57.1** | 29.6 |
| **Open-Source General and Reasoning MLLMs (32B)** | | | | | | | | |
| InternVL2.5-VL-38B | 71.9 | 49.4 | 31.8 | 32.0 | 67.5 | 46.0 | 57.6 | - |
| Qwen-2.5-VL-32B | 74.7 | 48.5 | 38.4 | 30.0 | 69.1 | 49.5 | 59.4 | 31.1 |
| InternVL2.5-38B-MPO | 73.8 | 46.5 | 32.3 | 25.6 | 66.2 | – | – | - |
| MM-Eureka-32B | 74.8 | 56.5 | 34.4 | 35.9 | 73.4 | [†]50.4 | [†]62.3 | [†]34.5 |
| ★ (Ours - **SRPO-32B**) | **78.5** | **58.9** | **39.6** | **38.5** | **76.4** | **51.3** | **66.1** | **38.2** |

including InternVL2.5-MPO [54], OpenVLThinker-7B [55], MM-Eureka-7B [56], VL-Rethinker-7B [15], R1-Onevision-7B [11], and R1-VL-7B [39]. We evaluate SRPO across three categories of multimodal reasoning benchmarks: mathematical reasoning (MathVista [1], MathVerse [2], MathVision [27], OlympiadBench [57], WeMath [58]), general reasoning (MMMU-Pro [3], MMMU [59], EMMA [60]), and cross-disciplinary reasoning (MMK12 [8]), covering physics, chemistry, and biology tasks.

**Implementation Setup.** For self-reflection cold-start SFT and subsequent RL training, Qwen2.5-VL-7B-Instruct and Qwen2.5-VL-32B-Instruct models are trained on 8 and 32 NVIDIA H100 GPUs, respectively. We adopt 1 epoch for SFT to avoid overfitting. During RL, we adopt the Open-RLHF framework [61], training for 3 epochs on 30K samples with rollout and training batch sizes set to 128 (8 rollouts per sample), a sampling temperature of 1.0, and Adam optimizer with a learning rate of $1 \times 10^{-6}$. For the reflection reward parameter $\alpha$, we set it to 0.1 to ensure training stability. Regarding the reflective brevity reward $f_{\text{len}}(L_{\text{response}})$, to discourage excessively verbose outputs, we define $T_{\text{target}}$ as 2× the length of the original response (i.e., reflection plus new reasoning equals the first think length), and set $T_{\text{max}}$ to 2.5× the original length (i.e., reflection plus new reasoning equals 1.5× the first think length). Additional hyper-parameter settings and detailed prompt configurations are provided in Appendix B.3 and Appendix B.4.

Table 2: Performance comparison across different disciplines in MMK12.

| Model | Math | Phys | Chem | Bio |
|---|---|---|---|---|
| **Closed Models** | | | | |
| Claude3.7 | 57.4 | 53.4 | 55.4 | 55.0 |
| GPT-4o | 55.8 | 41.2 | 47.0 | 55.4 |
| o1 | 81.6 | 68.8 | 71.4 | 74.0 |
| Gemini2 | 76.8 | 53.6 | 64.6 | 66.0 |
| **Open General MLLMs** | | | | |
| IntVL2.5-8B | 46.8 | 35.0 | 50.0 | 50.8 |
| Qwen-2.5-7B | 58.4 | 45.4 | 56.4 | 54.0 |
| IntVL2.5-38B | 61.6 | 49.8 | 60.4 | 60.0 |
| Qwen-2.5-32B | 71.6 | 59.4 | 69.6 | 66.6 |
| Qwen-2.5-72B | 75.6 | 64.8 | 69.6 | 72.0 |
| **Open Reasoning MLLMs** | | | | |
| IntVL2.5-8B-MPO | 26.6 | 25.0 | 42.4 | 44.0 |
| IntVL2.5-38B-MPO | 41.4 | 42.8 | 55.8 | 53.2 |
| R1-OneVision | 44.8 | 33.8 | 39.8 | 40.8 |
| MM-Eureka-7B | 71.2 | 56.2 | 65.2 | 65.2 |
| OpenVLThinker | 63.0 | 53.8 | 60.6 | 65.0 |
| MM-Eureka-32B | 74.6 | 62.0 | 75.4 | 76.8 |
| SRPO-7B | 75.3 | 60.6 | 70.3 | 69.5 |
| SRPO-32B | **77.5** | **64.2** | **77.5** | **79.2** |

## 4.2 Main Results of Benchmarks

**Multimodal General Reasoning.** We further evaluate our approach on general multimodal reasoning tasks to assess the effectiveness of our reflection-enhanced training strategy beyond mathematical

Table 3: Ablation study of SRPO-7B on RL training data size and self-reflection components.

| Model Components | RL Data Size | MathVista | MathVerse | MathVision | MMMU-Pro | Physics | Avg. |
|---|---|---|---|---|---|---|---|
| Qwen-2.5-VL-7B | - | 68.2 | 46.3 | 25.1 | 36.9 | 45.4 | 44.4 |
| + GRPO | 37K | 72.3 | 52.9 | 30.3 | 39.9 | 53.5 | 49.8 |
| ⋆ (Ours - **SRPO-7B**) | 37K | **75.8** | **55.8** | **32.9** | **42.3** | **60.6** | **53.5** |
| SRPO-7B | 15K | 74.5 | 54.9 | 32.2 | 41.4 | 60.1 | 52.6 |
| SRPO-7B | 5K | 73.7 | 53.6 | 31.2 | 40.3 | 57.7 | 51.3 |
| w/o *Self-Reflection SFT* | 37K | 74.2 | 53.3 | 30.3 | 39.7 | 58.6 | 51.2 |
| w/o *Self-Reflection RL* | 37K | 70.3 | 48.2 | 27.2 | 38.7 | 48.5 | 46.6 |
| - no Length Reward ($f_{\text{len}}(\cdot)$) | 37K | 75.3 | 56.2 | 32.4 | 41.7 | 60.1 | 53.1 |
| - no Effectiveness Reward ($I_{\text{eff}}$) | 37K | 73.9 | 54.7 | 31.6 | 40.9 | 58.8 | 52.0 |

reasoning. As shown in Table 1, SRPO-7B consistently outperforms existing open-source MLLMs on three general-domain benchmarks: MMMU-Pro, MMMU, and EMMA. Notably, compared to state-of-the-art closed-source reasoning models, SRPO-32B still demonstrates highly competitive performance, exceeding Gemini2-flash by 4.6 on the EMMA benchmark. These results underscore the broader generalizability of reflection-enhanced training in improving multimodal reasoning capabilities.

**Multimodal Mathematical Reasoning.** As presented in Table 1, SRPO achieves highly competitive performance on multiple mathematical reasoning benchmarks, even when compared to leading closed-source MLLMs. For instance, on the MathVista benchmark, SRPO obtains a score of 78.5%, trailing the widely acknowledged state-of-the-art model, OpenAI GPT-o1, by only 73.9%. Moreover, SRPO consistently outperforms open-source general multimodal baselines by a clear margin. Notably, when compared to state-of-the-art open-source reasoning models such as VL-Rethinker-7B and MM-Eureka-7B, SRPO demonstrates obvious advantages, even on complex, graduate-level reasoning datasets like OlympiadBench. These results strongly validate our claim that explicitly enhancing the model's self-reflection capabilities during both the SFT and RL stages positively contributes to improved complex reasoning performance.

**Cross-disciplinary Reasoning.** Beyond evaluating our model on widely-used multimodal mathematical and general reasoning benchmarks, we also investigate its capability for cross-disciplinary generalization to novel tasks not included in the training data, such as physics, chemistry, and biology. Results presented in Table 2 demonstrate that SRPO achieves superior cross-disciplinary reasoning performance, surpassing MM-Eureka-7B (trained solely via RL without self-reflection incentivization) by 5.1 points on Physics and OpenVLThinker-7B (SFT-enhanced reasoning) by 9.7 points on Chemistry. These findings highlight that integrating both reflection-enhanced SFT during the cold-start stage and employing a reflection-aware reward function during the RL stage significantly improves the model's generalization to previously unseen reasoning domains.

### 4.3 Ablation Study

**RL Train-Set Size.** We analyze SRPO's performance sensitivity to the RL training set size by sampling subsets of 15K and 7K from our original 37K dataset. As shown in Table 4.3, SRPO consistently improves with more data. Remarkably, even at 5K samples, SRPO significantly outperforms Qwen-2.5-VL-7B and standard GRPO, exceeding GRPO by 7.1 points on the Physics benchmark. Thus, enhancing self-reflection within RL efficiently boosts reasoning even under limited data.

**Effectiveness of Self-Reflection.** We further investigate individual self-reflection components within SRPO. Table 4.3 shows that removing Self-Reflection SFT notably reduces performance, yet still maintains a 5.1-point advantage over standard GRPO on Physics. Conversely, eliminating Self-Reflection RL yields minimal improvements over Qwen-2.5-VL-7B, indicating that reflection training solely in the SFT stage is insufficient. Hence, explicitly incentivizing reflection behavior during RL is essential for ehancing multimodal reasoning.

**Effectiveness of Reflection-Aware RL Components.** We also observe that omitting any specific Self-Reflection RL component can degrade performance, especially when the Effectiveness Reward ($I_{\text{eff}}$) is removed, resulting in a drop in average performance from 53.5 to 52.0. It indicates that the model critically relies on reward signals that explicitly evaluate the quality of reflective responses to achieve optimal reasoning. Similarly, reducing the Length Reward ($f_{\text{len}}$) also leads to a decline in reasoning performance, suggesting that overly redundant thinking steps can interfere with the model's accurate reasoning.

Table 4: Comparison between SRPO and GRPO with 2-Step Thinking.

| Methods | MathVista | MathVerse | MathVision | MMMU-Pro | Physics | Avg. |
|---|---|---|---|---|---|---|
| ⋆ (Ours - **SRPO-7B**) | **75.8** | **55.8** | **32.9** | **42.3** | **60.6** | **53.5** |
| GRPO | 72.3 | 52.9 | 30.3 | 39.9 | 53.5 | 49.8 |
| GRPO + 2-Step Thinking | 73.5 | 53.6 | 30.6 | 40.3 | 53.6 | 50.3 |

**Comparative Study of Reflective and Two-Step Thinking.** To assess the effectiveness of our proposed reflection pattern—comprising `thinking`, `reflection`, and `rethinking`—we compare it against a GRPO-based 2-step thinking paradigm, where the model generates two consecutive `<think>...</think>` steps. Training uses the task reward from Equation 5 and a relational reward inspired by Equation 8 that captures consistency between the two steps. As shown in Table 4, GRPO + 2-Step Thinking offers no significant gains over vanilla GRPO, except on MathVerse. In contrast, SRPO's explicit reflection on prior thinking substantially improves reasoning, underscoring the importance of combining Self-Reflection SFT with Reflection-aware RL.

**Disentangling the Effect of Reflection Format vs. Teacher Distillation** To examine whether the SFT improvement stems from teacher distillation or the proposed reflection structure, we conduct controlled experiments on the 7B model. All models share the same teacher (GPT-4-mini), dataset size (10k), optimizer, and number of training epochs. The only differences lie in the inclusion of `<reflection>` segments and whether reflection-aware RL is applied. The results are shown in Table 5. Adding explicit reflection segments in SFT yields an average gain of +1.3 over plain CoT SFT, indicating that structured reflection rather than teacher distillation contributes to better reasoning performance. When both models undergo RL, SRPO achieves an additional +2.1 gain by aligning rewards with reflection effectiveness, which highlights the benefit of reflection-aware reward design.

Table 5: **Ablation of Self-Reflection SFT on the 7B model.**

| Methods | MathVista | MathVerse | MathVision | MMMU-Pro | Physics | Avg. |
|---|---|---|---|---|---|---|
| Qwen-2.5-VL-7B | 68.2 | 46.3 | 25.1 | 36.9 | 45.4 | 44.4 |
| Plain-CoT SFT | 69.1 | 47.2 | 26.4 | 37.2 | 47.6 | 45.3 |
| Reflection-SFT (ours) | 70.3 | 48.2 | 27.2 | 38.7 | 48.5 | 46.6 |
| Plain-CoT SFT + GRPO | 73.6 | 54.2 | 30.6 | 40.6 | 58.0 | 51.4 |
| Reflection-SFT + Reflection-RL | **75.8** | **55.8** | **32.9** | **42.3** | **60.6** | **53.5** |

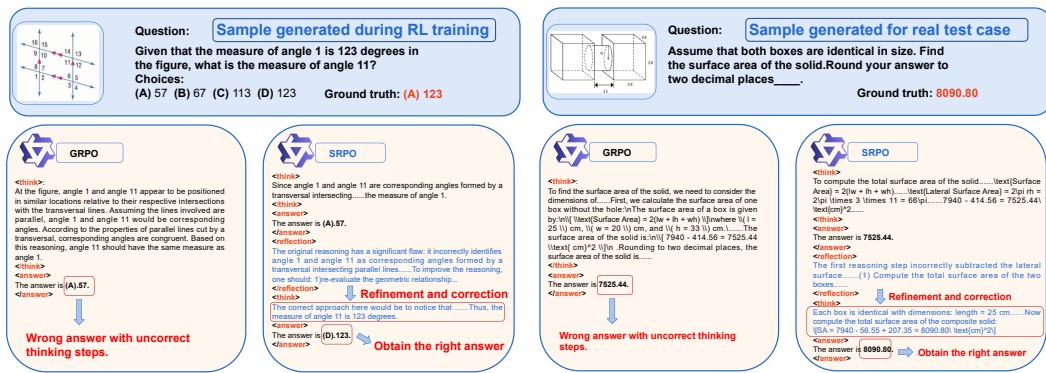

Figure 3: Generated samples in RL training (**left**) and generated samples in real test case (**right**).

## 4.4 Reasoning Qualitative Analysis

**Self-Reflection in RL Training.** Figure 3 The left part compares samples generated during RL training with SRPO and standard GRPO using Qwen-2.5-VL-7B, specifically highlighting the intermediate reasoning steps. We observe that SRPO explicitly guides the model to engage in effective self-reflection on its initial reasoning paths and answers. During the reflection process, SRPO corrects wrong reasoning steps and provides concise revisions, leading to refined final answers. In contrast, GRPO-generated samples typically contain reasoning steps without explicit reflective

corrections, rarely revising the initial reasoning paths. We provide the complete version of samples in Appendix B.5.

**Self-Reflection in Test Case.** Futhermore, figure 3 right part illustrates examples of reasoning outputs during inference. Responses generated by SRPO exhibit clear self-reflection patterns acquired during RL training, actively refining or correcting flawed reasoning steps and answers to improve overall accuracy. Conversely, models trained without self-reflection rarely adjust or rectify incorrect reasoning paths, resulting in persistent reasoning errors and reduced final performance.

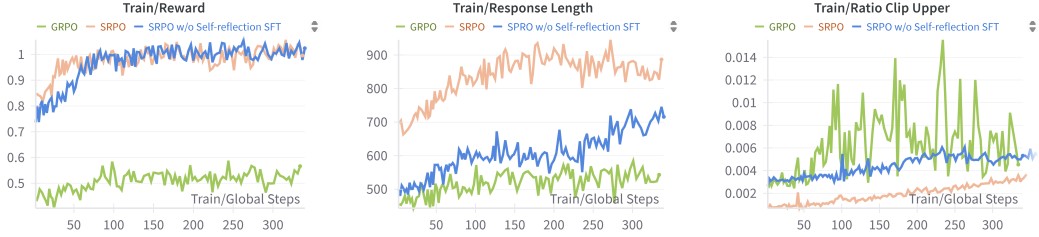

Figure 4: Training curves for SRPO and baselines: (a) training reward, (b) response length, and (c) upper clipping ratio.

### 4.5 Further Analysis

**RL Training Dynamics Analysis.** We analyze training dynamics to highlight SRPO's advantages (Figure 4). SRPO and SRPO w/o self-reflection SFT converge faster and outperform standard GRPO, illustrating that reflection-enhanced initialization accelerates reflection skill acquisition and improves reasoning. Moreover, SRPO consistently generates longer responses (Figure 4(b)), indicating effective early-stage reflection training from cold-start initialization. Interestingly, SRPO's lower, smoother *ratio clip upper* curve (Figure 4(c)) reflects stable policy updates, avoiding excessively large gradients or step sizes, confirming enhanced training consistency from reflection-based RL. More training visualizations are shown in Appendix B.6.

**Combining Self-Reflection with Alternative RL Methods.** To validate the generality of our self-reflection strategy, we incorporate it into PPO and DAPO algorithms within the OpenRLHF framework, following identical cold-start SFT initialization and evaluating checkpoints at 500 training steps. Figure 5 shows consistent improvements from self-reflection integration across all RL algorithms. Reflection-enhanced DAPO achieves performance comparable to SRPO, while SRPO slightly surpasses reflection-enhanced PPO. The result highlights the advantage of GRPO's group-based advantage estimation and reflection-oriented rewards over PPO's single-trajectory reward signals for effectively incentivizing self-reflection.

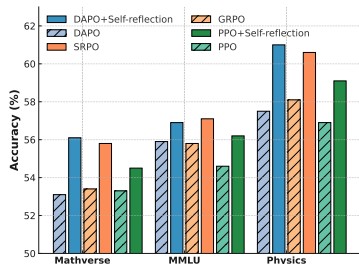

Figure 5: Performance of various RL methods with and without self-reflection.

## 5 Conclusion

In this paper, we introduced SRPO, a reflection-aware reinforcement learning framework designed to enhance multimodal reasoning capabilities in mutlimodal large language models. By systematically generating high-quality reflection-focused training data and employing a novel reward mechanism that explicitly incentivizes concise and effective self-reflection, our method successfully addresses the limitations of previous approaches, including insufficient data quality and lack of self-reflective behavior for refining response. Comprehensive experiments across multiple multimodal reasoning benchmarks demonstrated the significant effectiveness of *SRPO*, surpassing existing state-of-the-art models in both reasoning accuracy and reflection quality. Our results highlight the critical role of reflection-driven training strategies for robust multimodal reasoning.

## 6 Acknowledgment

We thank Xuehan Xiong, Kunchang Li and Qinghao Ye for their support in technical discussions related to this work. We also thank Faming Wu for his assistance in addressing infrastructure issues.

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

# NeurIPS Paper Checklist

1. **Claims**

   Question: Do the main claims made in the abstract and introduction accurately reflect the paper's contributions and scope?

   Answer: [Yes]

   Justification: We clearly present our contribution in abstract and introduction. And we list our contribution at the end of the introduction part.

   Guidelines:

   - The answer NA means that the abstract and introduction do not include the claims made in the paper.
   - The abstract and/or introduction should clearly state the claims made, including the contributions made in the paper and important assumptions and limitations. A No or NA answer to this question will not be perceived well by the reviewers.
   - The claims made should match theoretical and experimental results, and reflect how much the results can be expected to generalize to other settings.
   - It is fine to include aspirational goals as motivation as long as it is clear that these goals are not attained by the paper.

2. **Limitations**

   Question: Does the paper discuss the limitations of the work performed by the authors?

   Answer: [Yes]

   Justification: We discuss the limitations in Section A.1

   Guidelines:

   - The answer NA means that the paper has no limitation while the answer No means that the paper has limitations, but those are not discussed in the paper.
   - The authors are encouraged to create a separate "Limitations" section in their paper.
   - The paper should point out any strong assumptions and how robust the results are to violations of these assumptions (e.g., independence assumptions, noiseless settings, model well-specification, asymptotic approximations only holding locally). The authors should reflect on how these assumptions might be violated in practice and what the implications would be.
   - The authors should reflect on the scope of the claims made, e.g., if the approach was only tested on a few datasets or with a few runs. In general, empirical results often depend on implicit assumptions, which should be articulated.
   - The authors should reflect on the factors that influence the performance of the approach. For example, a facial recognition algorithm may perform poorly when image resolution is low or images are taken in low lighting. Or a speech-to-text system might not be used reliably to provide closed captions for online lectures because it fails to handle technical jargon.
   - The authors should discuss the computational efficiency of the proposed algorithms and how they scale with dataset size.
   - If applicable, the authors should discuss possible limitations of their approach to address problems of privacy and fairness.
   - While the authors might fear that complete honesty about limitations might be used by reviewers as grounds for rejection, a worse outcome might be that reviewers discover limitations that aren't acknowledged in the paper. The authors should use their best judgment and recognize that individual actions in favor of transparency play an important role in developing norms that preserve the integrity of the community. Reviewers will be specifically instructed to not penalize honesty concerning limitations.

3. **Theory assumptions and proofs**

   Question: For each theoretical result, does the paper provide the full set of assumptions and a complete (and correct) proof?

   Answer: [NA] .

Justification: As discussed in the introduction. This paper mainly focus on the dataset construction, learning algorithm design and empirical studies.

Guidelines:

- The answer NA means that the paper does not include theoretical results.
- All the theorems, formulas, and proofs in the paper should be numbered and cross-referenced.
- All assumptions should be clearly stated or referenced in the statement of any theorems.
- The proofs can either appear in the main paper or the supplemental material, but if they appear in the supplemental material, the authors are encouraged to provide a short proof sketch to provide intuition.
- Inversely, any informal proof provided in the core of the paper should be complemented by formal proofs provided in appendix or supplemental material.
- Theorems and Lemmas that the proof relies upon should be properly referenced.

4. **Experimental result reproducibility**

Question: Does the paper fully disclose all the information needed to reproduce the main experimental results of the paper to the extent that it affects the main claims and/or conclusions of the paper (regardless of whether the code and data are provided or not)?

Answer: [Yes]

Justification: We provide the algorithm formula in Section 3 and experiment setups in Section 4.1. We have outlined all details about our dataset (Appendix B.2), hyperparameters (Appendix B.3), and prompt templates (Appendix B.4) in Appendix B.

Guidelines:

- The answer NA means that the paper does not include experiments.
- If the paper includes experiments, a No answer to this question will not be perceived well by the reviewers: Making the paper reproducible is important, regardless of whether the code and data are provided or not.
- If the contribution is a dataset and/or model, the authors should describe the steps taken to make their results reproducible or verifiable.
- Depending on the contribution, reproducibility can be accomplished in various ways. For example, if the contribution is a novel architecture, describing the architecture fully might suffice, or if the contribution is a specific model and empirical evaluation, it may be necessary to either make it possible for others to replicate the model with the same dataset, or provide access to the model. In general. releasing code and data is often one good way to accomplish this, but reproducibility can also be provided via detailed instructions for how to replicate the results, access to a hosted model (e.g., in the case of a large language model), releasing of a model checkpoint, or other means that are appropriate to the research performed.
- While NeurIPS does not require releasing code, the conference does require all submissions to provide some reasonable avenue for reproducibility, which may depend on the nature of the contribution. For example
  (a) If the contribution is primarily a new algorithm, the paper should make it clear how to reproduce that algorithm.
  (b) If the contribution is primarily a new model architecture, the paper should describe the architecture clearly and fully.
  (c) If the contribution is a new model (e.g., a large language model), then there should either be a way to access this model for reproducing the results or a way to reproduce the model (e.g., with an open-source dataset or instructions for how to construct the dataset).
  (d) We recognize that reproducibility may be tricky in some cases, in which case authors are welcome to describe the particular way they provide for reproducibility. In the case of closed-source models, it may be that access to the model is limited in some way (e.g., to registered users), but it should be possible for other researchers to have some path to reproducing or verifying the results.

5. **Open access to data and code**

Question: Does the paper provide open access to the data and code, with sufficient instructions to faithfully reproduce the main experimental results, as described in supplemental material?

Answer: [Yes]

Justification: We provide experiment setup description in Section 4.1. We have outlined all details about our dataset (Appendix B.2), hyperparameters (Appendix B.3), and prompt templates (Appendix B.4) in Appendix B.

Guidelines:

- The answer NA means that paper does not include experiments requiring code.
- Please see the NeurIPS code and data submission guidelines (`https://nips.cc/public/guides/CodeSubmissionPolicy`) for more details.
- While we encourage the release of code and data, we understand that this might not be possible, so "No" is an acceptable answer. Papers cannot be rejected simply for not including code, unless this is central to the contribution (e.g., for a new open-source benchmark).
- The instructions should contain the exact command and environment needed to run to reproduce the results. See the NeurIPS code and data submission guidelines (`https://nips.cc/public/guides/CodeSubmissionPolicy`) for more details.
- The authors should provide instructions on data access and preparation, including how to access the raw data, preprocessed data, intermediate data, and generated data, etc.
- The authors should provide scripts to reproduce all experimental results for the new proposed method and baselines. If only a subset of experiments are reproducible, they should state which ones are omitted from the script and why.
- At submission time, to preserve anonymity, the authors should release anonymized versions (if applicable).
- Providing as much information as possible in supplemental material (appended to the paper) is recommended, but including URLs to data and code is permitted.

6. **Experimental setting/details**

Question: Does the paper specify all the training and test details (e.g., data splits, hyperparameters, how they were chosen, type of optimizer, etc.) necessary to understand the results?

Answer: [Yes]

Justification: We have presented the details of experiments in the main paper (Section 4.1) and Appendix B.

Guidelines:

- The answer NA means that the paper does not include experiments.
- The experimental setting should be presented in the core of the paper to a level of detail that is necessary to appreciate the results and make sense of them.
- The full details can be provided either with the code, in appendix, or as supplemental material.

7. **Experiment statistical significance**

Question: Does the paper report error bars suitably and correctly defined or other appropriate information about the statistical significance of the experiments?

Answer: [No]

Justification: All reinforcement learning experiments were performed with a single fixed random seed. For downstream tasks, we adhere to the official benchmark protocols and utilize GPT-4o as the standardized API-based evaluator to ensure consistency across all reported results.

Guidelines:

- The answer NA means that the paper does not include experiments.
- The authors should answer "Yes" if the results are accompanied by error bars, confidence intervals, or statistical significance tests, at least for the experiments that support the main claims of the paper.

- The factors of variability that the error bars are capturing should be clearly stated (for example, train/test split, initialization, random drawing of some parameter, or overall run with given experimental conditions).
- The method for calculating the error bars should be explained (closed form formula, call to a library function, bootstrap, etc.)
- The assumptions made should be given (e.g., Normally distributed errors).
- It should be clear whether the error bar is the standard deviation or the standard error of the mean.
- It is OK to report 1-sigma error bars, but one should state it. The authors should preferably report a 2-sigma error bar than state that they have a 96% CI, if the hypothesis of Normality of errors is not verified.
- For asymmetric distributions, the authors should be careful not to show in tables or figures symmetric error bars that would yield results that are out of range (e.g. negative error rates).
- If error bars are reported in tables or plots, The authors should explain in the text how they were calculated and reference the corresponding figures or tables in the text.

8. **Experiments compute resources**

Question: For each experiment, does the paper provide sufficient information on the computer resources (type of compute workers, memory, time of execution) needed to reproduce the experiments?

Answer: [Yes]

Justification: We provide implementation setup in 4.1 and include all details in Appendix B.3

Guidelines:

- The answer NA means that the paper does not include experiments.
- The paper should indicate the type of compute workers CPU or GPU, internal cluster, or cloud provider, including relevant memory and storage.
- The paper should provide the amount of compute required for each of the individual experimental runs as well as estimate the total compute.
- The paper should disclose whether the full research project required more compute than the experiments reported in the paper (e.g., preliminary or failed experiments that didn't make it into the paper).

9. **Code of ethics**

Question: Does the research conducted in the paper conform, in every respect, with the NeurIPS Code of Ethics https://neurips.cc/public/EthicsGuidelines?

Answer: [Yes]

Justification: We have carefully checked the code of ethics.

Guidelines:

- The answer NA means that the authors have not reviewed the NeurIPS Code of Ethics.
- If the authors answer No, they should explain the special circumstances that require a deviation from the Code of Ethics.
- The authors should make sure to preserve anonymity (e.g., if there is a special consideration due to laws or regulations in their jurisdiction).

10. **Broader impacts**

Question: Does the paper discuss both potential positive societal impacts and negative societal impacts of the work performed?

Answer: [Yes]

Justification: We discuss the broader impacts of the paper in Section A.2.

Guidelines:

- The answer NA means that there is no societal impact of the work performed.
- If the authors answer NA or No, they should explain why their work has no societal impact or why the paper does not address societal impact.

- Examples of negative societal impacts include potential malicious or unintended uses (e.g., disinformation, generating fake profiles, surveillance), fairness considerations (e.g., deployment of technologies that could make decisions that unfairly impact specific groups), privacy considerations, and security considerations.
- The conference expects that many papers will be foundational research and not tied to particular applications, let alone deployments. However, if there is a direct path to any negative applications, the authors should point it out. For example, it is legitimate to point out that an improvement in the quality of generative models could be used to generate deepfakes for disinformation. On the other hand, it is not needed to point out that a generic algorithm for optimizing neural networks could enable people to train models that generate Deepfakes faster.
- The authors should consider possible harms that could arise when the technology is being used as intended and functioning correctly, harms that could arise when the technology is being used as intended but gives incorrect results, and harms following from (intentional or unintentional) misuse of the technology.
- If there are negative societal impacts, the authors could also discuss possible mitigation strategies (e.g., gated release of models, providing defenses in addition to attacks, mechanisms for monitoring misuse, mechanisms to monitor how a system learns from feedback over time, improving the efficiency and accessibility of ML).

11. **Safeguards**

Question: Does the paper describe safeguards that have been put in place for responsible release of data or models that have a high risk for misuse (e.g., pretrained language models, image generators, or scraped datasets)?

Answer: [Yes]

Justification: We have clarified the source of our data (Appendix B.2), and the benchmark is intended solely for academic research purposes.

Guidelines:

- The answer NA means that the paper poses no such risks.
- Released models that have a high risk for misuse or dual-use should be released with necessary safeguards to allow for controlled use of the model, for example by requiring that users adhere to usage guidelines or restrictions to access the model or implementing safety filters.
- Datasets that have been scraped from the Internet could pose safety risks. The authors should describe how they avoided releasing unsafe images.
- We recognize that providing effective safeguards is challenging, and many papers do not require this, but we encourage authors to take this into account and make a best faith effort.

12. **Licenses for existing assets**

Question: Are the creators or original owners of assets (e.g., code, data, models), used in the paper, properly credited and are the license and terms of use explicitly mentioned and properly respected?

Answer: [Yes]

Justification: We have cited and notified the works such as Mulberry Dataset and MathV360K (Appendix B.2).

Guidelines:

- The answer NA means that the paper does not use existing assets.
- The authors should cite the original paper that produced the code package or dataset.
- The authors should state which version of the asset is used and, if possible, include a URL.
- The name of the license (e.g., CC-BY 4.0) should be included for each asset.
- For scraped data from a particular source (e.g., website), the copyright and terms of service of that source should be provided.

- If assets are released, the license, copyright information, and terms of use in the package should be provided. For popular datasets, `paperswithcode.com/datasets` has curated licenses for some datasets. Their licensing guide can help determine the license of a dataset.
- For existing datasets that are re-packaged, both the original license and the license of the derived asset (if it has changed) should be provided.
- If this information is not available online, the authors are encouraged to reach out to the asset's creators.

13. **New assets**

    Question: Are new assets introduced in the paper well documented and is the documentation provided alongside the assets?

    Answer: [Yes]

    Justification: We explain our data construction pipeline in Section 3.1.1.

    Guidelines:
    - The answer NA means that the paper does not release new assets.
    - Researchers should communicate the details of the dataset/code/model as part of their submissions via structured templates. This includes details about training, license, limitations, etc.
    - The paper should discuss whether and how consent was obtained from people whose asset is used.
    - At submission time, remember to anonymize your assets (if applicable). You can either create an anonymized URL or include an anonymized zip file.

14. **Crowdsourcing and research with human subjects**

    Question: For crowdsourcing experiments and research with human subjects, does the paper include the full text of instructions given to participants and screenshots, if applicable, as well as details about compensation (if any)?

    Answer: [NA]

    Justification: We don't include crowdsourcing or any research with human subjects in this paper.

    Guidelines:
    - The answer NA means that the paper does not involve crowdsourcing nor research with human subjects.
    - Including this information in the supplemental material is fine, but if the main contribution of the paper involves human subjects, then as much detail as possible should be included in the main paper.
    - According to the NeurIPS Code of Ethics, workers involved in data collection, curation, or other labor should be paid at least the minimum wage in the country of the data collector.

15. **Institutional review board (IRB) approvals or equivalent for research with human subjects**

    Question: Does the paper describe potential risks incurred by study participants, whether such risks were disclosed to the subjects, and whether Institutional Review Board (IRB) approvals (or an equivalent approval/review based on the requirements of your country or institution) were obtained?

    Answer: [NA]

    Justification: This paper does not involve crowdsourcing nor research with human subjects.

    Guidelines:
    - The answer NA means that the paper does not involve crowdsourcing nor research with human subjects.
    - Depending on the country in which research is conducted, IRB approval (or equivalent) may be required for any human subjects research. If you obtained IRB approval, you should clearly state this in the paper.

- We recognize that the procedures for this may vary significantly between institutions and locations, and we expect authors to adhere to the NeurIPS Code of Ethics and the guidelines for their institution.
- For initial submissions, do not include any information that would break anonymity (if applicable), such as the institution conducting the review.

16. **Declaration of LLM usage**

Question: Does the paper describe the usage of LLMs if it is an important, original, or non-standard component of the core methods in this research? Note that if the LLM is used only for writing, editing, or formatting purposes and does not impact the core methodology, scientific rigorousness, or originality of the research, declaration is not required.

Answer: [Yes]

Justification: We use LLM to generate and refine our dataset sample (Section 3.1.1. We use LLM results as baselines to compare our proposed reflective reinforcement learning approach (Section 4.1). We adapt our reflective reinforcement learning approach on Qwen2.5-VL (Section 4.1.

Guidelines:

- The answer NA means that the core method development in this research does not involve LLMs as any important, original, or non-standard components.
- Please refer to our LLM policy (https://neurips.cc/Conferences/2025/LLM) for what should or should not be described.

17. **Claims**

Question: Do the main claims made in the abstract and introduction accurately reflect the paper's contributions and scope?

Answer: [Yes]

Justification: We clearly present our contribution in abstract and introduction. And we list our contribution at the end of the introduction part.

Guidelines:

- The answer NA means that the abstract and introduction do not include the claims made in the paper.
- The abstract and/or introduction should clearly state the claims made, including the contributions made in the paper and important assumptions and limitations. A No or NA answer to this question will not be perceived well by the reviewers.
- The claims made should match theoretical and experimental results, and reflect how much the results can be expected to generalize to other settings.
- It is fine to include aspirational goals as motivation as long as it is clear that these goals are not attained by the paper.

18. **Limitations**

Question: Does the paper discuss the limitations of the work performed by the authors?

Answer: [Yes]

Justification: We discuss the limitations in Section A.1

Guidelines:

- The answer NA means that the paper has no limitation while the answer No means that the paper has limitations, but those are not discussed in the paper.
- The authors are encouraged to create a separate "Limitations" section in their paper.
- The paper should point out any strong assumptions and how robust the results are to violations of these assumptions (e.g., independence assumptions, noiseless settings, model well-specification, asymptotic approximations only holding locally). The authors should reflect on how these assumptions might be violated in practice and what the implications would be.
- The authors should reflect on the scope of the claims made, e.g., if the approach was only tested on a few datasets or with a few runs. In general, empirical results often depend on implicit assumptions, which should be articulated.

- The authors should reflect on the factors that influence the performance of the approach. For example, a facial recognition algorithm may perform poorly when image resolution is low or images are taken in low lighting. Or a speech-to-text system might not be used reliably to provide closed captions for online lectures because it fails to handle technical jargon.
- The authors should discuss the computational efficiency of the proposed algorithms and how they scale with dataset size.
- If applicable, the authors should discuss possible limitations of their approach to address problems of privacy and fairness.
- While the authors might fear that complete honesty about limitations might be used by reviewers as grounds for rejection, a worse outcome might be that reviewers discover limitations that aren't acknowledged in the paper. The authors should use their best judgment and recognize that individual actions in favor of transparency play an important role in developing norms that preserve the integrity of the community. Reviewers will be specifically instructed to not penalize honesty concerning limitations.

19. **Theory Assumptions and Proofs**

    Question: For each theoretical result, does the paper provide the full set of assumptions and a complete (and correct) proof?

    Answer: [NA]

    Justification: As discussed in the introduction. This paper mainly focus on the dataset construction, learning algorithm design and empirical studies.

    Guidelines:

    - The answer NA means that the paper does not include theoretical results.
    - All the theorems, formulas, and proofs in the paper should be numbered and cross-referenced.
    - All assumptions should be clearly stated or referenced in the statement of any theorems.
    - The proofs can either appear in the main paper or the supplemental material, but if they appear in the supplemental material, the authors are encouraged to provide a short proof sketch to provide intuition.
    - Inversely, any informal proof provided in the core of the paper should be complemented by formal proofs provided in appendix or supplemental material.
    - Theorems and Lemmas that the proof relies upon should be properly referenced.

20. **Experimental Result Reproducibility**

    Question: Does the paper fully disclose all the information needed to reproduce the main experimental results of the paper to the extent that it affects the main claims and/or conclusions of the paper (regardless of whether the code and data are provided or not)?

    Answer: [Yes]

    Justification: We provide the algorithm formula in Section 3 and experiment setups in Section 4.1. We have outlined all details about our dataset, hyperparameters, and prompt templates in Appendix B.

    Guidelines:

    - The answer NA means that the paper does not include experiments.
    - If the paper includes experiments, a No answer to this question will not be perceived well by the reviewers: Making the paper reproducible is important, regardless of whether the code and data are provided or not.
    - If the contribution is a dataset and/or model, the authors should describe the steps taken to make their results reproducible or verifiable.
    - Depending on the contribution, reproducibility can be accomplished in various ways. For example, if the contribution is a novel architecture, describing the architecture fully might suffice, or if the contribution is a specific model and empirical evaluation, it may be necessary to either make it possible for others to replicate the model with the same dataset, or provide access to the model. In general. releasing code and data is often one good way to accomplish this, but reproducibility can also be provided via detailed

instructions for how to replicate the results, access to a hosted model (e.g., in the case of a large language model), releasing of a model checkpoint, or other means that are appropriate to the research performed.

- While NeurIPS does not require releasing code, the conference does require all submissions to provide some reasonable avenue for reproducibility, which may depend on the nature of the contribution. For example

  (a) If the contribution is primarily a new algorithm, the paper should make it clear how to reproduce that algorithm.

  (b) If the contribution is primarily a new model architecture, the paper should describe the architecture clearly and fully.

  (c) If the contribution is a new model (e.g., a large language model), then there should either be a way to access this model for reproducing the results or a way to reproduce the model (e.g., with an open-source dataset or instructions for how to construct the dataset).

  (d) We recognize that reproducibility may be tricky in some cases, in which case authors are welcome to describe the particular way they provide for reproducibility. In the case of closed-source models, it may be that access to the model is limited in some way (e.g., to registered users), but it should be possible for other researchers to have some path to reproducing or verifying the results.

21. **Open access to data and code**

Question: Does the paper provide open access to the data and code, with sufficient instructions to faithfully reproduce the main experimental results, as described in supplemental material?

Answer: [Yes]

Justification: We provide experiment setup description in Section 4.1. We have outlined all details about our dataset (Appendix B.2), hyperparameters (Appendix B.3), and prompt templates (Appendix B.4) in Appendix B.

Guidelines:

- The answer NA means that paper does not include experiments requiring code.
- Please see the NeurIPS code and data submission guidelines (`https://nips.cc/public/guides/CodeSubmissionPolicy`) for more details.
- While we encourage the release of code and data, we understand that this might not be possible, so "No" is an acceptable answer. Papers cannot be rejected simply for not including code, unless this is central to the contribution (e.g., for a new open-source benchmark).
- The instructions should contain the exact command and environment needed to run to reproduce the results. See the NeurIPS code and data submission guidelines (`https://nips.cc/public/guides/CodeSubmissionPolicy`) for more details.
- The authors should provide instructions on data access and preparation, including how to access the raw data, preprocessed data, intermediate data, and generated data, etc.
- The authors should provide scripts to reproduce all experimental results for the new proposed method and baselines. If only a subset of experiments are reproducible, they should state which ones are omitted from the script and why.
- At submission time, to preserve anonymity, the authors should release anonymized versions (if applicable).
- Providing as much information as possible in supplemental material (appended to the paper) is recommended, but including URLs to data and code is permitted.

22. **Experimental Setting/Details**

Question: Does the paper specify all the training and test details (e.g., data splits, hyperparameters, how they were chosen, type of optimizer, etc.) necessary to understand the results?

Answer: [Yes]

Justification: We have presented the details of experiments in the main paper (Section 4.1) and Appendix B.

Guidelines:

- The answer NA means that the paper does not include experiments.
- The experimental setting should be presented in the core of the paper to a level of detail that is necessary to appreciate the results and make sense of them.
- The full details can be provided either with the code, in appendix, or as supplemental material.

23. **Experiment Statistical Significance**

Question: Does the paper report error bars suitably and correctly defined or other appropriate information about the statistical significance of the experiments?

Answer: [NA]

Justification:

Guidelines:

- The answer NA means that the paper does not include experiments.
- The authors should answer "Yes" if the results are accompanied by error bars, confidence intervals, or statistical significance tests, at least for the experiments that support the main claims of the paper.
- The factors of variability that the error bars are capturing should be clearly stated (for example, train/test split, initialization, random drawing of some parameter, or overall run with given experimental conditions).
- The method for calculating the error bars should be explained (closed form formula, call to a library function, bootstrap, etc.)
- The assumptions made should be given (e.g., Normally distributed errors).
- It should be clear whether the error bar is the standard deviation or the standard error of the mean.
- It is OK to report 1-sigma error bars, but one should state it. The authors should preferably report a 2-sigma error bar than state that they have a 96% CI, if the hypothesis of Normality of errors is not verified.
- For asymmetric distributions, the authors should be careful not to show in tables or figures symmetric error bars that would yield results that are out of range (e.g. negative error rates).
- If error bars are reported in tables or plots, The authors should explain in the text how they were calculated and reference the corresponding figures or tables in the text.

24. **Experiments Compute Resources**

Question: For each experiment, does the paper provide sufficient information on the computer resources (type of compute workers, memory, time of execution) needed to reproduce the experiments?

Answer: [Yes]

Justification: We provide implementation setup in 4.1 and include all details in Appendix B.3

Guidelines:

- The answer NA means that the paper does not include experiments.
- The paper should indicate the type of compute workers CPU or GPU, internal cluster, or cloud provider, including relevant memory and storage.
- The paper should provide the amount of compute required for each of the individual experimental runs as well as estimate the total compute.
- The paper should disclose whether the full research project required more compute than the experiments reported in the paper (e.g., preliminary or failed experiments that didn't make it into the paper).

25. **Code Of Ethics**

Question: Does the research conducted in the paper conform, in every respect, with the NeurIPS Code of Ethics https://neurips.cc/public/EthicsGuidelines?

Answer: [Yes]

Justification: We have carefully checked the code of ethics.

Guidelines:

- The answer NA means that the authors have not reviewed the NeurIPS Code of Ethics.
- If the authors answer No, they should explain the special circumstances that require a deviation from the Code of Ethics.
- The authors should make sure to preserve anonymity (e.g., if there is a special consideration due to laws or regulations in their jurisdiction).

26. **Broader Impacts**

Question: Does the paper discuss both potential positive societal impacts and negative societal impacts of the work performed?

Answer: [Yes]

Justification: We discuss the broader impacts of the paper in Section A.2.

Guidelines:

- The answer NA means that there is no societal impact of the work performed.
- If the authors answer NA or No, they should explain why their work has no societal impact or why the paper does not address societal impact.
- Examples of negative societal impacts include potential malicious or unintended uses (e.g., disinformation, generating fake profiles, surveillance), fairness considerations (e.g., deployment of technologies that could make decisions that unfairly impact specific groups), privacy considerations, and security considerations.
- The conference expects that many papers will be foundational research and not tied to particular applications, let alone deployments. However, if there is a direct path to any negative applications, the authors should point it out. For example, it is legitimate to point out that an improvement in the quality of generative models could be used to generate deepfakes for disinformation. On the other hand, it is not needed to point out that a generic algorithm for optimizing neural networks could enable people to train models that generate Deepfakes faster.
- The authors should consider possible harms that could arise when the technology is being used as intended and functioning correctly, harms that could arise when the technology is being used as intended but gives incorrect results, and harms following from (intentional or unintentional) misuse of the technology.
- If there are negative societal impacts, the authors could also discuss possible mitigation strategies (e.g., gated release of models, providing defenses in addition to attacks, mechanisms for monitoring misuse, mechanisms to monitor how a system learns from feedback over time, improving the efficiency and accessibility of ML).

27. **Safeguards**

Question: Does the paper describe safeguards that have been put in place for responsible release of data or models that have a high risk for misuse (e.g., pretrained language models, image generators, or scraped datasets)?

Answer: [Yes]

Justification: We have clarified the source of our data (Appendix B.2), and the benchmark is intended solely for academic research purposes.

Guidelines:

- The answer NA means that the paper poses no such risks.
- Released models that have a high risk for misuse or dual-use should be released with necessary safeguards to allow for controlled use of the model, for example by requiring that users adhere to usage guidelines or restrictions to access the model or implementing safety filters.
- Datasets that have been scraped from the Internet could pose safety risks. The authors should describe how they avoided releasing unsafe images.
- We recognize that providing effective safeguards is challenging, and many papers do not require this, but we encourage authors to take this into account and make a best faith effort.

28. **Licenses for existing assets**

    Question: Are the creators or original owners of assets (e.g., code, data, models), used in the paper, properly credited and are the license and terms of use explicitly mentioned and properly respected?

    Answer: [Yes]

    Justification: We have cited and notified the works such as Mulberry Dataset and MathV360K (Appendix B.2).

    Guidelines:

    - The answer NA means that the paper does not use existing assets.
    - The authors should cite the original paper that produced the code package or dataset.
    - The authors should state which version of the asset is used and, if possible, include a URL.
    - The name of the license (e.g., CC-BY 4.0) should be included for each asset.
    - For scraped data from a particular source (e.g., website), the copyright and terms of service of that source should be provided.
    - If assets are released, the license, copyright information, and terms of use in the package should be provided. For popular datasets, `paperswithcode.com/datasets` has curated licenses for some datasets. Their licensing guide can help determine the license of a dataset.
    - For existing datasets that are re-packaged, both the original license and the license of the derived asset (if it has changed) should be provided.
    - If this information is not available online, the authors are encouraged to reach out to the asset's creators.

29. **New Assets**

    Question: Are new assets introduced in the paper well documented and is the documentation provided alongside the assets?

    Answer: [Yes]

    Justification: We explain our data construction pipeline in Section 3.1.1.

    Guidelines:

    - The answer NA means that the paper does not release new assets.
    - Researchers should communicate the details of the dataset/code/model as part of their submissions via structured templates. This includes details about training, license, limitations, etc.
    - The paper should discuss whether and how consent was obtained from people whose asset is used.
    - At submission time, remember to anonymize your assets (if applicable). You can either create an anonymized URL or include an anonymized zip file.

30. **Crowdsourcing and Research with Human Subjects**

    Question: For crowdsourcing experiments and research with human subjects, does the paper include the full text of instructions given to participants and screenshots, if applicable, as well as details about compensation (if any)?

    Answer: [NA]

    Justification: We don't include crowdsourcing or any research with human subjects in this paper.

    Guidelines:

    - The answer NA means that the paper does not involve crowdsourcing nor research with human subjects.
    - Including this information in the supplemental material is fine, but if the main contribution of the paper involves human subjects, then as much detail as possible should be included in the main paper.

- According to the NeurIPS Code of Ethics, workers involved in data collection, curation, or other labor should be paid at least the minimum wage in the country of the data collector.

31. **Institutional Review Board (IRB) Approvals or Equivalent for Research with Human Subjects**

   Question: Does the paper describe potential risks incurred by study participants, whether such risks were disclosed to the subjects, and whether Institutional Review Board (IRB) approvals (or an equivalent approval/review based on the requirements of your country or institution) were obtained?

   Answer: [NA]

   Justification: This paper does not involve crowdsourcing nor research with human subjects.

   Guidelines:

   - The answer NA means that the paper does not involve crowdsourcing nor research with human subjects.
   - Depending on the country in which research is conducted, IRB approval (or equivalent) may be required for any human subjects research. If you obtained IRB approval, you should clearly state this in the paper.
   - We recognize that the procedures for this may vary significantly between institutions and locations, and we expect authors to adhere to the NeurIPS Code of Ethics and the guidelines for their institution.
   - For initial submissions, do not include any information that would break anonymity (if applicable), such as the institution conducting the review.

## A    Limitation and Social Impacts

### A.1    Limitation

Our experiments primarily evaluated the effectiveness of SRPO on dense MLLMs at 7B and 32B scales, without conducting scaling experiments on Mixture-of-Experts (MoE) [62, 63] or diffusion LM [64–66] architectures. Additionally, the reinforcement learning training data utilized in our experiments were selected exclusively from publicly available multimodal reasoning datasets, without exploration of larger-scale commercial reasoning datasets. Extending our method to MoE-based models and incorporating larger-scale RL training datasets remain promising avenues for future work, potentially leading to even broader improvements in multimodal reasoning capabilities.

### A.2    Social Impacts

Our work offers clear positive contributions by significantly enhancing MLLMs' capabilities in complex reasoning tasks. These advances can enable more accurate and reliable AI assistance in education, scientific discovery, and decision-making scenarios, ultimately contributing to broader accessibility of high-quality reasoning support. However, improved reasoning capabilities also carry potential risks, such as generating more convincing yet inaccurate or biased content if models reflect on misleading data. Consequently, careful curation of training datasets and explicit efforts in mitigating potential biases remain essential. Additionally, deploying advanced multimodal reasoning models without adequate safeguards might inadvertently reinforce existing societal inequalities if access to these advanced technologies remains restricted. Overall, responsible and transparent use of these enhanced multimodal reasoning frameworks is crucial to ensure positive societal outcomes.

## B    Appendix

In this appendix, we provide supplementary details and extended analyses supporting the main findings presented in our paper. Specifically, we describe the datasets employed for self-reflection SFT and RL training (§B.2), present detailed hyper-parameter settings used throughout our experiments (§B.3), and provide explicit prompt templates for self-reflection SFT, RL training, and evaluation (§B.4). Additionally, we illustrate training dynamics, including convergence trends and key metrics

during the reinforcement learning phase (§B.6). Finally, we offer qualitative analyses through representative samples generated during RL training and inference, demonstrating the effectiveness of our reflection-based strategies (§B.5).

## B.1 Further Experiment

**Sensitivity to the Reflection Length Reward Coefficient $\alpha$.** To evaluate the robustness of SRPO to the reflection-length reward coefficient, we additionally trained models with $\alpha = 0.05$ and $\alpha = 0.3$, using the same setup as in the main experiments. The results are shown in Table 6. All experiments were performed under identical optimization and data settings, with only the length-reward coefficient varied. Results indicate that $\alpha = 0.05$ performs similarly to $\alpha = 0.1$, while $\alpha = 0.3$ slightly degrades accuracy. This suggests that over-emphasizing length reward weakens the reflection signal, whereas values near $0.1$ offer a balanced trade-off, helping to control response length and stabilize training with limited impact on accuracy.

Table 6: Sensitivity of SRPO to the reflection length reward coefficient $\alpha$.

| Ratio | MathVerse | MMMU-Pro | Physics | Avg. |
|---|---|---|---|---|
| $\alpha$=0.05 | 55.7 | 41.9 | 61.1 | 52.9 |
| $\alpha$=0.1 (Ours) | **55.8** | **42.3** | **60.6** | **53.5** |
| $\alpha$=0.3 | 55.1 | 42.0 | 59.7 | 52.2 |

**Qualitative & quantitative reflection analysis.** To quantitatively assess reflection effectiveness, we conducted both human and LLM-based evaluations on SRPO-7B's generated reflections.

**a. Human Expert Evaluation.** We randomly sampled 100 MathVista test questions (answered by SRPO-7B). Two senior PhD students specializing in NLP and LLMs independently rated each reflection on a 0–3 scale: 3 = highly effective, 2 = partially effective, 1 = redundant, and 0 = detrimental. We also measured the *Wrong Answer Fix Rate*, i.e., how often an initially incorrect answer was corrected after reflection. The results are summarized in Table 7.

Table 7: **Human Expert Evaluation of Reflection Quality.**

| Metric | Human Expert 1 | Human Expert 2 |
|---|---|---|
| Effective Reflection Rate (score $\geq$ 2) | 73% | 69% |
| Redundancy Rate (score = 1) | 9% | 11% |
| Detrimental Rate (score = 0) | 3% | 1% |
| Wrong Answer Fix Rate | 39% | 39% |

Seventy percent of reflections were judged to be effective. Among 100 questions, the initial solution was incorrect in 33 cases, of which 13 were corrected after reflection, resulting in a 39% wrong-answer fix rate.

**b. LLM-as-a-Judge (GPT-4o) Evaluation.** Each reflection was further scored by GPT-4o on four 0–5 dimensions—logical flaws, missing assumptions, clarity, and actionable suggestions—following the prompt format described in Appendix B.3. The average reflection quality is shown in Table 8.

Table 8: **LLM-as-a-Judge Evaluation (GPT-4o).**

| Model | Logic | Missing | Clarity | Suggestions | Avg. Quality |
|---|---|---|---|---|---|
| SRPO-7B (Ours) | 4.1 | 3.9 | 4.0 | 3.6 | **3.9** |

These results confirm that SRPO produces high-quality reflections from both human and LLM perspectives, validating the effectiveness of its self-reflection design.

### Training and Inference Efficiency

We report the measured wall-clock time for each SRPO training stage under both 7B and 32B settings using the OpenRLHF framework. All settings share the same data size and optimization schedule as

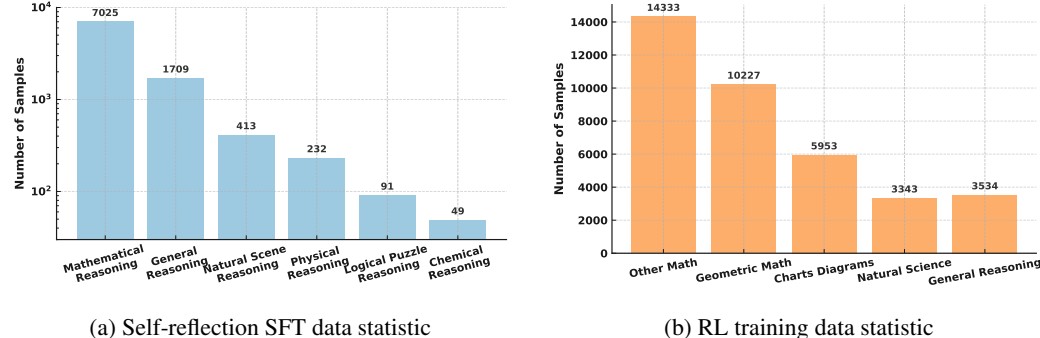

(a) Self-reflection SFT data statistic          (b) RL training data statistic

Figure 6: Statistics of reasoning-type distribution in different stages of training.

in the main experiments; only the model size and stage differ. Table 9 summarizes training efficiency, and Table 10 summarizes inference efficiency on the MathVista test-mini split ( 1k samples) using a single H100-80G GPU with vLLM. SRPO requires only modest additional wall-time compared with GRPO, while delivering substantially better reasoning quality in the main results.

Table 9: Training efficiency overview. Wall-clock time for each stage under 7B and 32B settings.

| Training Stage | GPUs | Wall-Time | Notes |
|---|---|---|---|
| Reflection-SFT (7B) | 8×H100 | 3.5 hours | 10k samples, 1 epoch |
| Reflection-RL (7B, SRPO) | 8×H100 | 31.2 hours | 37k samples, 500 steps |
| GRPO-RL (7B) | 8×H100 | 25.8 hours | 37k samples, 500 steps |
| Reflection-SFT (32B) | 4 × 8×H100 | 4.7 hours | 10k samples, 1 epoch |
| Reflection-RL (32B, SRPO) | 4 × 8×H100 | 45.1 hours | 37k samples, 500 steps |

Table 10: Inference efficiency overview. Latency on MathVista test-mini using a single H100-80G with vLLM.

| Inference Stage | GPUs | Wall-Time | Notes |
|---|---|---|---|
| GRPO-RL (7B) | 1×H100 | 45.6 min | 1k test samples, avg response = 355.4 tokens |
| SRPO (7B) | 1×H100 | 60.5 min | 1k test samples, avg response = 502.3 tokens |

## B.2 Training Dataset

**Self-reflection SFT Dataset.** We primarily select samples from three established multimodal reasoning datasets:

- **Mulberry Dataset (260K)** [24]: A multimodal reasoning dataset enriched through collective Monte Carlo tree search, specifically designed to enhance reflection and reasoning capabilities of multimodal LLMs. It features diverse reasoning problems requiring explicit cognitive processes.
- **MathV360K** [40]: The dataset focuses on mathematical reasoning for multimodal LLMs. It systematically bootstraps multimodal reasoning by constructing high-quality mathematical reasoning prompts paired with visual contexts.
- **LLaVA-CoT Dataset (100K)** [25]: A vision-language reasoning dataset explicitly designed for chain-of-thought (CoT) prompting. It consists of multimodal problems that encourage step-by-step logical reasoning aligned with visual inputs.

From these datasets, we randomly sample 100K data points. For each sampled multimodal problem, we apply a specialized CoT-generation template, feeding both the visual inputs and associated questions to two pretrained models: Qwen-2.5-VL-7B-Instruct and Qwen-2.5-VL-32B-Instruct [28], respectively, to generate corresponding reasoning steps. Subsequently, we use the DeepSeek-V3 [67] API to assess the quality of these generated reasoning paths, selecting a high-quality subset of 10K samples containing approximately 30% correctly solved and 70% incorrectly solved reasoning paths.

Next, leveraging these selected CoT samples and their associated ground truths, we utilize GPT-o4-mini [7] with a specifically designed self-reflection generation prompt. This approach yields concise, meaningful self-reflective feedback regarding the generated reasoning steps. Finally, we structure these samples into our self-reflection enhanced SFT dataset following the proposed Self-reflection SFT template.

**Self-reflection RL Dataset.** We curate our reinforcement learning (RL) dataset by selectively sampling from several multimodal reasoning benchmarks, each featuring distinct reasoning characteristics and data modalities. These datasets include:

- **ScienceQA** [41]: Contains 21K multimodal science questions, encouraging explicit reasoning chains through visual contexts, textual explanations, and multiple-choice tasks.

- **Geometric Math QA (GeoQA)** [42]: Consists of 5K geometry-focused multimodal problems designed to evaluate numerical reasoning over geometric concepts.

- **ChartQA** [43]: Provides 9.6K questions for visual and logical reasoning based on diverse chart types such as bar, line, and pie charts.

- **DVQA** [44]: Includes 3.5K questions requiring comprehensive reasoning to interpret data visualizations effectively.

- **AI2D** [45]: Features 5K diagram-based science questions aimed at evaluating visual and conceptual understanding through diagrams.

- **MATH** [46]: Comprises 12.5K challenging mathematical problems across various difficulty levels, extensively utilized to measure mathematical reasoning capabilities.

- **Virgo** [47]: Offers around 10K multimodal reasoning examples intended to emulate the reasoning complexity of state-of-the-art models like OpenAI O1.

- **R1-OneVision** [11]: Contains 5K multimodal reasoning instances specifically designed for cross-modal reasoning and generalization.

- **MMK12** [8]: Provides a curated set of around 12K multimodal problems that involve rigorous rule-based reasoning across diverse domains.

- **PhyX** [48]: Comprises 3K carefully designed multimodal physics questions spanning thermodynamics, electromagnetism, mechanics, modern physics, optics, and acoustics, aimed at evaluating physical reasoning capabilities in realistic visual contexts.

By systematically combining and sampling from these datasets, we obtain a comprehensive self-reflection RL dataset containing high-quality multimodal reasoning instances for our experiments.

## B.3 Hyper-parameters

In the reinforcement learning phase of SRPO, we highlight several critical hyper-parameters: we set both the rollout and training batch sizes to 128, generating 8 samples per prompt to ensure response diversity. Sampling is performed with a temperature of 1.0. The learning rate is fixed at $1 \times 10^{-6}$ using the Adam optimizer with parameter offloading enabled, and training is conducted using bf16 mixed-precision. We adopt group-normalized advantage estimation (specific to GRPO) to stabilize training and utilize the "k3" KL divergence estimator for controlled policy updates. Additionally, we freeze the visual encoder parameters during training, enable gradient checkpointing and flash attention for memory efficiency, and perform accuracy filtering (retaining samples with accuracy scores between 0.1 and 0.9) to maintain data quality throughout the RL process.

## B.4 Prompt Template

---

**Prompt Template for CoT Generation**

**System:** "You are a precise AI assistant and must strictly follow the following rules:

- First reason step-by-step, and wrap the thought process in `<think>` tags.
- The final answer must be wrapped in `<answer>` tags.
- Formatting requirements:
    - Choice answers must be uppercase letters (A/B/C/D).
    - Fill-in-the-blank answers should be digits.
- DO NOT EXPLAIN ANYTHING IN `<answer>`.
- You must provide both `<think>` and `<answer>`.
- Please strictly follow the formatting requirements and do not add any extra content!

**User:** `[type: "text", content: question,`
`type: "image_url", image_url: image_url]`

---

**Prompt Template for Self-Reflection Generation**

**System:** "You are a helpful math reasoning assistant. Think carefully. Output only JSON."

**User:**

You are an expert visual reasoning assistant. Your task is to reflect on the quality of a chain-of-thought (CoT) reasoning given for a visual question. The goal is to **improve** the CoT by identifying weaknesses and offering suggestions for refinement.

Please follow this structure strictly:

1. `"reflection"`: Provide a detailed critique of the original CoT, pointing out:
    - Logical flaws or inconsistencies
    - Missing assumptions or information
    - Any correct reasoning that could be made clearer or more robust
    - Suggestions for improving the reasoning process

Only return a valid JSON object with a `"reflection"` field.

```
-- Input --
Question: {query}
Image: {image_url}
Original Chain of Thought: {cot}
Predicted Answer (Based on CoT): {answer}
Correct Answer (ground truth): {ground truth}
```

---

## Prompt Template for Self-reflection SFT

**System:** You are a reasoning expert. Given an image and a question, please generate two rounds of step-by-step reasoning:

- First, provide your initial chain of thought and answer.
- Then reflect on it.
- Finally, based on your reflection, give your final reasoning and answer.

**User:**

```
{
  "messages": [
    {
      "role": "system",
      "content": "You are a reasoning expert. Given an image and a question,
      please generate two rounds of step-by-step reasoning: First,
      provide your initial chain of thought and answer. Then reflect
      on it, and finally, based on your reflection,
      give your final reasoning and answer."
    },
    {
      "role": "user",
      "content": "Question: <question>{query}</question>\nImage: <image>"
    },
    {
      "role": "assistant",
      "content": "<think>{cot}</think>\n<answer>{answer}</answer>
      \n<reflection>{reflection}</reflection>
      \n<answer>{ground_truth}</answer>"
    }
  ],
  "images": ["{image_url}"]
}
```

## Prompt Template for Self-reflection RL

**System:** Solve the user's question step by step.
First, think about the reasoning process internally and write it inside `<think>...</think>` tags. Then, provide the first answer in LaTeX format wrapped with `$...$`, and the final result must be enclosed with `\boxed{}`. Wrap this answer inside `<answer>...</answer>` tags.
After that, perform a critical self-reflection on the previous reasoning and answer, writing the reflection inside `<reflection>...</reflection>` tags.
Then, based on the reflection, generate a new reasoning process and a new answer:

- The new reasoning is again placed inside `<think>...</think>`.
- The new answer is written inside `<answer>...</answer>` and uses LaTeX `$...$` with `\boxed{}` for the final output.

Make sure both reasoning steps are clear and detailed. Even if the final answer does not change, the second reasoning must incorporate improvements based on the reflection.

**Format Example:**

```
<think> Since $1+1=2$, so the answer is $2$. </think>
<answer> The answer is $\boxed{2}$. </answer>
<reflection> The reasoning is correct but too brief;
I could have explained the addition more explicitly. </reflection>
<think> Adding $1$ and $1$ together results in $2$ because
$1$ plus $1$ means taking one and adding another one, leading to $2$. </think>
<answer> The answer is $\boxed{2}$. </answer>
```

All reasoning, answer, and reflection steps must be included **without omission** and in the exact sequence: `<think>...</think>` `<answer>...</answer>` `<reflection>...</reflection>` `<think>...</think>` `<answer>...</answer>`.

## B.5 Generated Samples Analysis

In this section, we present generated samples during RL training (Sample 1 and Sample 2) and from real test cases (Sample 3 and Sample 4). In the training examples (Sample 1 and 2), we observe that Qwen-2.5-VL-7B, under our proposed SRPO framework, exhibits the ability to perform self-reflection and subsequently revise its reasoning steps. Similarly, in downstream evaluation tasks, the model demonstrates reflective adjustments when answering incorrectly, and when answering correctly, it further refines and simplifies its reasoning through self-reflection.

---

**Generated Samples in RL Training of SRPO (Sample 1)**

**Question**: Given that the measure of angle 1 is 123 degrees in the figure, what is the measure of angle 11?

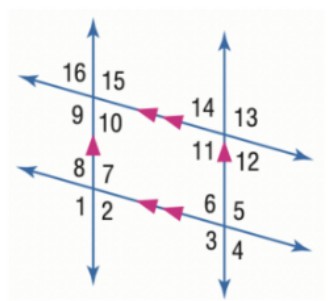

**Choices**: (A) 57    (B) 67    (C) 113    (D) 123    **Ground truth: (D) 123**

**<think>**
Since angle 1 and angle 11 are corresponding angles formed by a transversal intersecting two parallel lines, they are congruent. Therefore, the measure of angle 11 is the same as the measure of angle 1.
**</think>**
**<Answer>**
The answer is 57 (**Incorrect due to flawed assumptions**)
**</Answer>**
**<reflection>**
The original reasoning has a significant flaw: it incorrectly identifies angle 1 and angle 11 as corresponding angles formed by a transversal intersecting parallel lines. This assumption is unfounded given the image description, which does not specify that the lines are parallel. Moreover, the reasoning does not consider the possibility that angle 11 might be obtained from a different relationship, such as a vertical angle or an alternate interior angle formed by a non-parallel configuration. To improve the reasoning, one should: 1) analyze the given visual elements more carefully rather than relying on assumptions about parallel lines; 2) re-evaluate the geometric relationships present in the figure, including the possibility of vertical angles or other angle pairs that could be equal; and 3) include intermediate steps or calculations that confirm why the measure of angle 11 should be different from angle 1. Clarifying these points would lead to a more robust and accurate conclusion.
**</reflection>**
**<think>** (**Generate the right answer after effective self-reflection**)
The correct approach here would be to notice that if angle 1 and angle 11 were somehow related in a way that the sum of their measures was a specific constant (for example, if they were supplementary or complementary in context), we could deduce the measure of angle 11 by subtracting from 180 degrees (since supplementary angles add up to 180 degrees). But since the problem does not specify this relationship, and considering the answer choices, the most logical deduction is that angle 11 is the same as angle 1 due to a specific geometric configuration not detailed in the problem statement.Thus, the measure of angle 11 is 123°.
**</think>**
**<Answer>** The answer is 123
**</Answer>**

**Question**: Based on the upper image, what task is the robotic arm most likely performing?

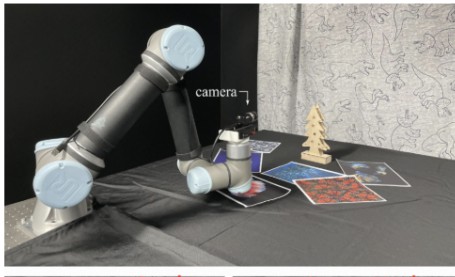
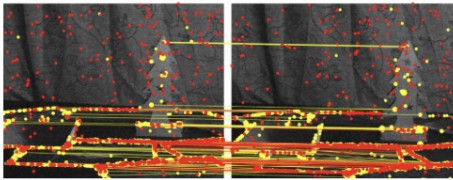

**Options**: (A) Sorting playing cards based on a predefined algorithm.
(B) Performing a magic trick with the cards.
(C) Playing a card game against a human opponent.
(D) Conducting an experiment on object recognition.     **Ground truth: (D)**

**<think>**
The image shows a robotic arm with a camera mounted on it, which suggests that it is involved in a task that requires visual input and processing. The presence of playing cards on the table and the arm's position above them indicate that the robot might be interacting with the cards in some way. Given the context of the setup, the most logical task for the robotic arm would be to sort the playing cards based on a predefined algorithm, as this is a common application for robotic arms equipped with cameras for visual recognition.
**</think>**

**<answer>** The answer is $\boxed{A}$ . (**Incorrect**)
**</answer>**

**<reflection>**
The original reasoning is overall correct in observing the camera and the cards, but it overlooks deeper experimental context. The red and yellow point clouds and structured laser-like patterns shown in the lower part of the image are indicative of 3D perception or object localization, which are typical in object recognition experiments. Additionally, there's no evidence of sorting mechanisms or card repositioning. The reflection should thus account for the experimental nature of the setup, making option D more consistent with the evidence provided.
**</reflection>**

**<think>** (**Generate the right answer after effective self-reflection**)
Let's rethink, the image shows a robotic arm with a camera, indicating a task that involves visual data collection or processing. The presence of playing cards on the table suggests an interaction with objects, and the alignment of the dots and lines might represent data points collected for object recognition or spatial analysis. Given these observations, the most likely task for the robotic arm is related to object recognition, where the camera would be used to identify and analyze the cards' features.
**</think>**

**<answer>**
The answer is $\boxed{D}$ .
**</answer>**

**Question**: A farmer wants to build a fence around the entire perimeter of his land, as shown in the diagram. The fencing costs £37 per metre. The lengths of $EF$ and $CD$ are $x$ metres and $y$ metres respectively. At £37 per metre, how much will it cost him to build the fence along the entire perimeter?
**Ground truth: 777**

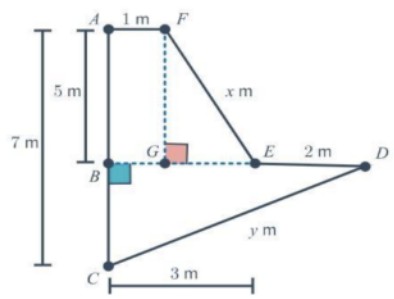

**\<think\>**
To determine the cost of building the fence around the entire perimeter of the land, we first need to find the total perimeter of the land. The perimeter is the sum of the lengths of all the sides of the shapes forming the land.
Given: - The height of the rectangle ABGF is 5 m. - The width of the rectangle ABGF is 1 m. - The height of the rectangle BCDE is 7 m. - The width of the rectangle BCDE is 2 m. - The length of EF is $x$ m. - The length of CD is $y$ m.
The perimeter of the land is the sum of the lengths of all the sides:

$$\text{Perimeter} = AB + BG + GE + EF + FD + DC + CE + EB$$

From the diagram:

$$\text{Perimeter} = 5 + 1 + 2 + x + 1 + 2 + 3 + 1 = 15 + x$$

Fencing cost is:

$$\boxed{555 + 37x}$$

**\</think\>**
**\<answer\>**
The answer is $\boxed{555 + 37x}$.  (**Incorrect**)
**\</answer\>**
**\<reflection\>**
The initial reasoning misidentified the structure of the perimeter by incorrectly including internal segments such as $BG$, $GE$, and $EB$. These are auxiliary construction lines used solely to form right triangles for computing the diagonal lengths $EF$ and $CD$, not actual parts of the outer boundary of the land that require fencing. This led to an overestimation of the total perimeter.
To correct the reasoning, one must focus strictly on the visual geometry and trace only the external contour of the land—specifically, the true fencing path comprises segments $AB$, $AF$, $EF$, $ED$, $DC$, and $CA$. The diagonal lengths $EF$ and $CD$ should be computed using the Pythagorean theorem based on right triangle constructions derived from the diagram. This corrected understanding yields the true perimeter and thus the accurate total fencing cost.
**\</reflection\> \<think\>**  (**Generate the right answer after effective self-reflection**)
Tracing only the external boundary gives segments

$$AB = 5, \ AF = 1, \ FE = x, \ ED = 2, \ DC = y, \ CB = 2.$$

Therefore the total perimeter is

$$5 + 1 + x + 2 + y + 2 = 10 + (x + y).$$

From the diagram's length proportions (and the single "11 m" label spanning the two slanted edges), we see directly that

$$x + y = 11.$$

Hence the full perimeter is $10 + 11 = 21$ m, and the fencing cost is

$$37 \times 21 = \boxed{777}.$$

**\</think\>**
**\<answer\>**
The answer is $\boxed{777}$.
**\</answer\>**

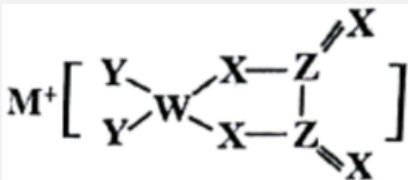

**Question**: M, W, X, Y, and Z are main-group elements in the same period. The number of valence electrons in X is three times the number of electrons in the penultimate shell of W. The element R is a key player in inorganic non-metal materials. The structure of the compound formed by M, W, X, Y, and Z is shown. Which of the following statements is incorrect?

**Choices**:

- **A.** Elemental M is usually stored under kerosene.

- **B.** The oxide of R dissolves in aqueous solution of Y's hydride.

- **C.** Every atom in the anion shown satisfies an octet.

- **D.** The boiling points of the simplest hydrides of X, Y, Z follow $X > Y > Z$.

**\<think\>**
Determine the period: all elements are in period 3. W's penultimate shell contains 8 electrons, so $X$ has $3 \times 8 = 24$ valence electrons; therefore, $X = \mathrm{Cl}$. Using known oxidation states: M = Na, W = Mg, Y = Al, Z = P. R is most likely Si. Now evaluate the options: A. Sodium is not stored under kerosene but under oil; therefore, this statement is incorrect.
B, C, and D are chemically consistent. Therefore, the incorrect statement is $\boxed{A}$.
**\</think\>**

**\<answer\>**
The answer is $\boxed{A}$.
**\</answer\>**

**\<reflection\>** **Reflect on the redundancy of the prior reasoning and generate a simplified version.**
The initial reasoning is correct but can be simplified. Instead of step-by-step element identification, we can streamline the logic into three clear steps: (1) From W's inner shell (8 electrons), deduce that X has 24 valence electrons, hence $X = \mathrm{Cl}$.
(2) Identify M = Na, W = Mg, Y = Al, Z = P from standard main-group patterns. (3) Eliminate each option: only A fails (Na is not stored under kerosene). This improved structure yields the same answer more directly.
**\</reflection\>**

**\<think\>**

From periodic positions and common valences: M = Na, W = Mg, Y = Al, Z = P. Only statement A is chemically incorrect. Answer: $\boxed{A}$.
**\</think\>**

**\<answer\>**
The answer is $\boxed{A}$.
**\</answer\>**

## B.6 Training Dynamics

We provide additional visualizations of RL training curves, including total response length, correct and incorrect response lengths, ratio clip lower, policy loss, and accuracy reward. Several key observations can be made: First, due to the explicit emphasis on self-reflection during training, SRPO consistently generates longer total responses and exhibits notably greater growth in response length compared to GRPO and SRPO without self-reflection. This is attributable to the model's active engagement in self-reflection and subsequent correction of prior reasoning steps. Additionally, SRPO consistently achieves higher accuracy reward values than baselines, confirming that reinforcement of reflective reasoning effectively enhances the model's reasoning capabilities. Furthermore, from the ratio clip lower and policy loss curves, we observe that SRPO—whether employing self-reflection in

both SFT and RL phases or solely in the RL phase—maintains stable clip lower values consistently below 0.005. This indicates that the integration of self-reflection contributes to stable policy updates with moderate gradient adjustments throughout training.

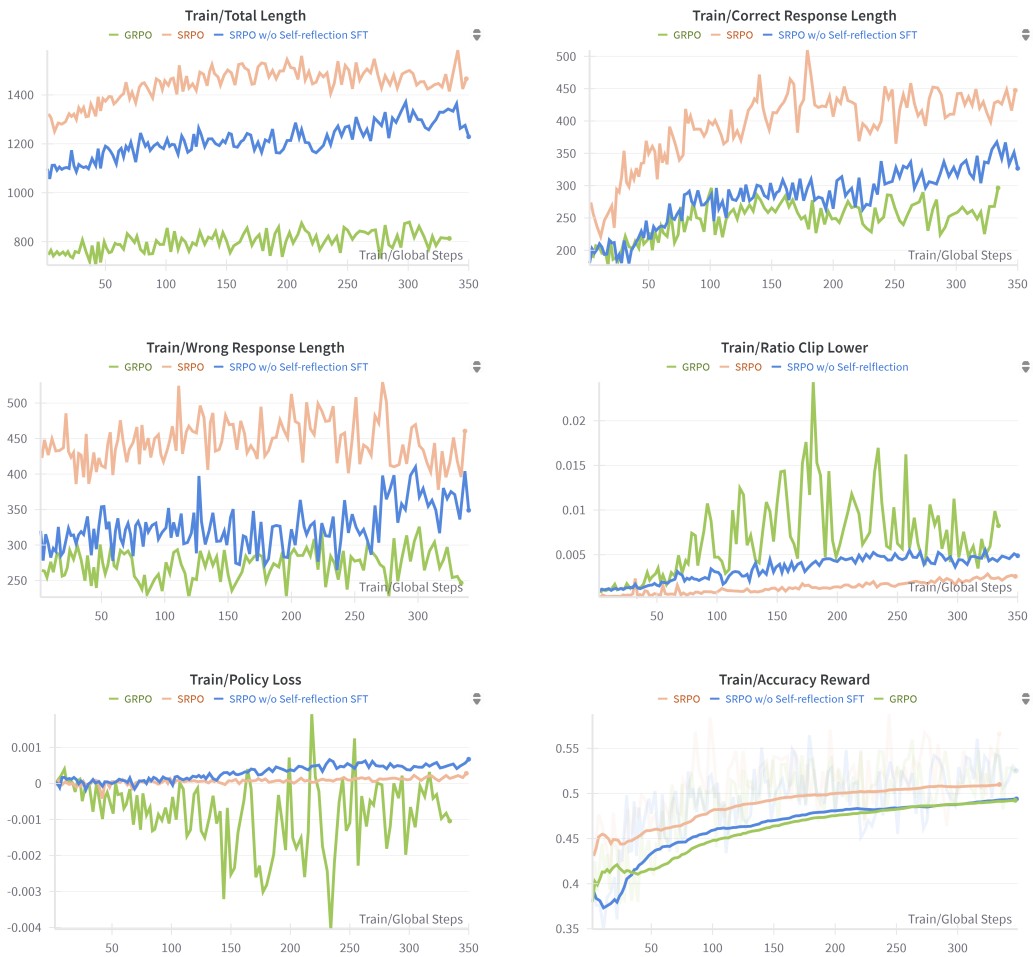

Figure 7: More training dynamics of SRPO, GRPO, and SRPO w/o Self-reflection SFT.

