# OpenReview forum: "SRPO: Enhancing Multimodal LLM Reasoning via Reflection-Aware Reinforcement Learning"
_NeurIPS.cc/2025/Conference — NeurIPS 2025 poster_

### Official Review · Reviewer_wAHj · 2025-06-16

**Clarity:** 3
**Significance:** 3
**Originality:** 3
**Rating:** 4
**Confidence:** 4

**Summary:**

The paper introduces SRPO, a two-stage training framework designed to enhance the reasoning capabilities of multimodal large language models (MLLMs) through explicit self-reflection. Motivated by the limitations of existing MLLMs—particularly their tendency to produce redundant or incorrect reasoning steps due to local token-level generation—the authors propose incorporating reflection into both the supervised fine-tuning (SFT) and reinforcement learning (RL) stages.
In the first stage, the model generates reflections by comparing its predictions with ground truth answers, using a strong MLLM (e.g., GPT-o4-mini) to guide the creation of high-quality reflection data. This reflection-augmented data is then used to fine-tune the policy model, encouraging both reasoning and self-correction behavior.
In the second stage, the authors introduce a reflection-aware reinforcement learning strategy built on the Group Relative Policy Optimization (GRPO) algorithm. A novel reward function explicitly rewards concise and task-relevant reflections while penalizing verbosity or redundancy, helping to mitigate reward hacking and promote meaningful reasoning.
Empirical results on multiple multimodal reasoning benchmarks (MathVista, MathVision, MMMU-Pro) using Qwen-2.5-VL models demonstrate that SRPO consistently outperforms existing methods in reasoning accuracy, reflection quality, and generalization. The paper’s main contributions include the reflection-based SFT pipeline, a novel reward design for reflective RL, and comprehensive evaluation validating the effectiveness of the approach.

**Questions:**

1. Can the authors provide more detailed qualitative and quantitative analyses of the self-reflection outputs? For example, how often does reflection successfully correct reasoning errors, and what types of errors are most effectively addressed?
2. Can the authors elaborate on possible strategies to reduce training and inference overhead associated with the reflection steps?
3. Have the authors tested the SRPO framework on reasoning tasks beyond STEM fields, such as legal, medical, or commonsense reasoning benchmarks? If not, can they discuss the potential or challenges for domain transfer?
4. The paper notes SRPO still trails some closed-source models on key benchmarks. Can the authors identify specific technical factors or limitations causing this gap? Are there targeted improvements or ablation studies planned to close this performance gap?

**Ethical Concerns:**

["NO or VERY MINOR ethics concerns only"]

**Final Justification:**

Thank you for the author's reply. After consideration, I have decided to keep my original score.

**Limitations:**

yes

**Quality:**

3

**Strengths And Weaknesses:**

Strengths
1.Effective Self-Reflection Mechanism
The proposed SRPO framework introduces self-reflection in both supervised fine-tuning (SFT) and reinforcement learning (RL) stages, enabling the model to identify and revise incorrect reasoning paths, leading to substantial improvements in complex multimodal reasoning.
2.Strong Benchmark Performance
SRPO demonstrates competitive or superior performance across a wide range of multimodal reasoning benchmarks, including MathVista, EMMA, and MMK12, outperforming existing open-source models and approaching the performance of top closed-source models.
3.Excellent Cross-Disciplinary Generalization
The model shows strong generalization to unseen scientific disciplines such as physics, chemistry, and biology, significantly outperforming other open-source reasoning-focused models like MM-Eureka and OpenVLThinker.
4.Generalizable to Multiple RL Algorithms
The reflection-enhanced training strategy is not limited to a single RL method. It improves performance consistently across PPO, DAPO, and GRPO, showing the versatility of the proposed mechanism.
5.Stable and Efficient Training
SRPO exhibits faster convergence and more stable policy updates during training, as evidenced by smoother learning curves and longer, more coherent generated responses during early RL stages.
6.Robust Under Low-Resource Settings
Even with reduced training data (e.g., 5K samples), SRPO maintains strong performance, indicating high data efficiency and robustness in low-resource scenarios.
Weaknesses
1.High Computational Cost
Training SRPO requires substantial computational resources (e.g., up to 32 H100 GPUs) and involves multi-stage optimization, making replication and deployment challenging for smaller research groups.
2.Inference Overhead Due to Reflection
The self-reflection mechanism introduces additional reasoning and revision steps, which can significantly increase inference time and limit applicability in latency-sensitive real-world settings.
3.Still Trails Closed-Source SOTA in Some Benchmarks
While close, SRPO still underperforms leading proprietary models like GPT-o1 and Gemini2 on certain tasks, such as MathVista, suggesting room for further optimization.
4.Limited Evidence of Domain Transferability
The current evaluation focuses primarily on STEM-related tasks. It remains unclear whether the self-reflection approach generalizes well to reasoning tasks in other domains such as legal or medical fields.
5.Lack of In-Depth Analysis of Reflection Quality
Although self-reflection improves accuracy, the paper lacks a thorough qualitative or quantitative analysis of the reflection content itself, which limits interpretability and deeper understanding of model behavior.

---

> ### Author Rebuttal · Authors · 2025-07-30
>
> ### **Response to reviewer wAHj**
>
> Thank you for the encouraging and constructive feedback. We appreciate your recognition of SRPO’s strengths, including its effective self-reflection design, strong benchmark performance, and broad generalization ability. Below, we address the reviewer’s comments and questions in detail.
>
> > **Response to Weakness 1. (Training Cost)**:
>
> We thank the reviewer for raising the concern on training efficiency. Below we report actual wall-clock times for SRPO under 7B and 32B settings using OpenRLHF [1]:
>
> #### **Table a.Training Efficiency Overview**
>
> | Training Stage               | GPUs         | Wall-Time  | Notes                        |
> |---------------------|--------------|------------|------------------------------|
> | Reflection-SFT (7B) | 8×H100       | 3.5 hours  | 10k samples, 1 epoch         |
> | Reflection-RL (7B)  | 8×H100       | 31.2 hours   | 37k samples, 500 steps       |
> | GRPO-RL (7B)  | 8×H100       | 25.8 hours   | 37k samples, 500 steps       |
> | Reflection-SFT (32B)| 4×8×H100     | 4.7 hours  | 10k samples, 1 epoch         |
> | Reflection-RL (32B) | 4×8×H100     | 45.1 hours   | 37k samples, 500 steps       |
>
> As shown in Table a, SRPO remains cost-effective despite its two-stage design. The SFT phase completes within 5 hours even for 32B models, and the RL stage is comparable to GRPO. The reflection component adds only moderate overhead, keeping training practical for academic-scale labs.
>
> As shown in Table b, to speed up training, we used 4 nodes of 8×H100 GPUs. Importantly, SRPO is fully runnable on a single 8×H100 (80G) setup under the verl framework [2] using BF16. While this increases training time, it supports smaller hardware setups. We will open-source both single-node and multi-node training pipelines in the revised version.
>
> #### **Table b. Hardware Requirements on verl framework.**
>
> | Method               | Bits  | 1.5B     | 3B      | 7B       | 32B       |
> |----------------------|-------|----------|---------|----------|-----------|
> | GRPO Full Fine-Tuning | AMP   | 2×24GB   | 4×40GB  | 8×40GB   | 16×80GB   |
> | GRPO Full Fine-Tuning | BF16  | 1×24GB   | 1×40GB  | 4×40GB   | 8×80GB    |
>
> [1] OpenRLHF: An Easy-to-use, Scalable and High-performance RLHF Framework
>
> [2] HybridFlow: A Flexible and Efficient RLHF Framework
>
> > **Response to Weakness 2.  (Inference Overhead)**:
>
> We thank the reviewer for highlighting the inference latency concern. On the MathVista test-mini set (~1k samples), we benchmarked **SRPO-7B** against **Qwen2.5-VL-7B + GRPO** using a single H100 (80G) and vLLM. SRPO incurs a 1.33× increase in wall-time and 1.41× more tokens due to the reflection step. However, our brevity-aware reward **f_len**​ effectively controls verbosity. As shown in Table 3, this modest overhead results in +3.5 accuracy gain on MathVista and +3.7 on average across benchmarks, highlighting a favorable trade-off between latency and reasoning quality.
>
>
> #### **Table c. Inference Efficient Overview**
>
> | Inference Stage               | GPUs         | Wall-Time  | Notes                        |
> |---------------------|--------------|------------|------------------------------|
> | GRPO-RL (7B)  | 1×H100     | 45.6min  | 1K test samples, avg response=355.4 tokens   |
> | SRPO (7B)  | 1×H100     | 60.5min  | 1K test samples, avg response=502.3 tokens   |
>
> > **Response to Question 1 and weakness 5. (Qualitative & quantitative reflection analysis)**:
>
> We thank the reviewer for the constructive suggestion regarding reflection quality analysis.
>
> Due to space limitations in the main text, Figure 3 shows abbreviated samples. We provide full reflection traces from both RL training and downstream evaluation in Appendix B.4, which demonstrate the model’s ability to self-reflect and revise its reasoning steps.
>
> Following your suggestion, we further conducted a human evaluation of reflection quality:
>
> **1. Human Expert Study for Reflection Quality:**
>
> We randomly selected 100 MathVista test samples (answered by SRPO-7B). Two additional senior PhD students (NLP/LLMs) rated each reflection on a 0–3 scale:
> • 3 = Highly effective • 2 = Partially effective • 1 = Redundant • 0 = Detrimental
> We also measured the Wrong Answer Fix Rate: how often initially incorrect answers were corrected after reflection.
> #### Table d. Human Expert Evaluation
> | Metric                          | Human Expert 1 | Human Expert 2 |
> |--------------------------------|----------------|----------------|
> | Effective Reflection Rate (score ≥ 2) | 73%           | 69%           |
> | Redundancy Rate (score = 1)           | 9%            | 11%           |
> | Detrimental Rate (score = 0)          | 3%            | 1%            |
> | Wrong Answer Fix Rate                 | 39%           | 39%           |
>
> This shows that 70% of reflections are helpful. Out of 100 questions, the initial solution was incorrect for 33 cases. Among them, 13 were corrected after reflection and a second attempt, resulting in a wrong answer fix rate of 39%.
>
> **2. LLM-as-a-Judge Evaluation (GPT‑4o) for Reflection Quality:**
>
> Each reflection was scored on 4 dimensions (0–5 scale): logical flaws, missing assumptions, clarity, and actionable suggestions following **B.3 Prompt Template. Prompt Template for Self-Reflection Generation**.
>
> #### **Table e. LLM-as-a-Judge Evaluation (GPT‑4o)**
> | Model     | Logic | Missing | Clarity | Suggestions | Avg Quality |
> |-----------|--------|----------|---------|--------------|----------|
> | SRPO-7B   | 4.1    | 3.9      | 4.0     | 3.6          | **3.9** |
>
> These results confirm high-quality reflections from both human and LLM perspectives.
>
> **3. Error types analysis**
>
> We analyze the cases from MathVista where the initial answers were incorrect and compare those corrected after reflection with those that remained incorrect. The key findings are summarized below.
>
> - **Most Effectively Corrected Errors**: SRPO was most effective at fixing mid-level algorithmic slips, such as arithmetic miscounts, sign errors, step-wise inconsistencies, coordinate read-offs, and option selection mistakes. These issues are internal to the reasoning chain and can be self-verified and revised by the model during reflection.
>
> - **Least Effectively Corrected Errors**: Reflection struggled with errors caused by visual misinterpretation, OCR noise, or misuse of domain knowledge (e.g.,math theorems). These errors require external grounding, such as reprocessing the image or consulting factual resources, which the current reflection mechanism cannot access.
>
> Therefore, Self-reflection serves as a reliable post-hoc validator for internal logical consistency, particularly for symbolic reasoning errors. For failures rooted in perception or knowledge, future work may integrate visual re-inspection or retrieval modules to extend SRPO’s repair capacity.
>
> > **Response to Question 2. (Possible Strategies for Reducing Training and Inference Overhead with Reflection):**
>
> We thank the reviewer for the insightful question. To reduce training and inference cost, we propose two practical extensions:
>
> - **Dynamic Reflection Triggering:**  We introduce a lightweight classifier to decide whether to trigger reflection based on input difficulty. Trained with a reward that balances accuracy and latency, this allows easy queries to skip reflection entirely, reducing average inference time.
>
> - **Length-Aware Reflection Control:**   We extend SRPO with a learned reflection-length planner that predicts a token budget based on task complexity. This reduces unnecessary generation during both training and inference, while preserving reasoning quality.
>
> These strategies make SRPO more efficient by dynamically allocating compute based on task demands.
>
> > **Response to Question 3. (Domain Transfer Beyond STEM):**
>
> We thank the reviewer for this important question. Although our current evaluation focuses on STEM tasks, prior work suggests that reflective RL frameworks like SRPO can generalize to other domains. Huan et al. [1] show that reasoning skills learned from math tasks via SFT+RL transfer well to medical and general knowledge QA. Similarly, Liu et al. [2] demonstrate that reasoning trained on general-domain text can transfer to multimodal and medical tasks, even without multimodal supervision. These results suggest SRPO’s mechanisms are broadly applicable given suitable textual reasoning data. We plan to evaluate on legal, medical, and commonsense benchmarks in future work.
>
> [1] Does Math Reasoning Improve General LLM Capabilities?
>
> [2] X-REASONER: Towards Generalizable Reasoning Across Modalities and Domains
>
> > **Response to Question 4 and Weakness 3. (Performance Gap with Closed-Source Models):**
>
> We appreciate the reviewer’s observation. While SRPO-32B outperforms many open-source and some closed-source baselines (e.g., on MathVista), it may trail larger proprietary systems like Seed1.5-VL-T. This gap is likely due to differences in capacity (e.g., >500B MoE vs. our 32B dense), scale of training data, and auxiliary tools like OCR and retrieval modules. To address this, we are (i) scaling SRPO to larger models, (ii) collecting higher-quality reflection data, potentially from human experts, and (iii) integrating high-res visual inputs and tool-augmented reflection. These are ongoing and will be included in future works.

---

> > ### Author Response · Authors · 2025-08-04
> >
> > Dear Reviewer wAHj, We are deeply grateful for the attention and care you've given to our work. Understanding the importance of thorough feedback, we're here to address any queries or points of ambiguity regarding our response. Please feel free to reach out with any further questions.

---

### Official Review · Reviewer_J4Dj · 2025-06-22

**Clarity:** 3
**Significance:** 3
**Originality:** 3
**Rating:** 4
**Confidence:** 4

**Summary:**

This paper aims to address the limitations of multimodal large language models (MLLMs) in handling complex reasoning tasks that require explicit self-reflection and self-correction. To tackle this issue, the authors propose SRPO, a two-stage reflection-aware reinforcement learning framework. In the first stage, a high-quality reflection-focused dataset is constructed with the help of an advanced MLLM. The second stage introduces a novel reward mechanism within the GRPO framework to promote concise and effective self-reflection while avoiding redundancy.

**Questions:**

1. Could author further explain the necessity of process of “first solution-> reflection-> second solution”? It may not be the same as the process of human reflection, which tend not to give an answer first, but rather focus on reflection on each intermediate step.
2. This paper calculate R_{accuracy} between answer and the first solution. Are the role of second solution just for reflection reward?
3. Could you please explain how you determine the optimal length T_{target}? I don’t find explanation about this in the paper.
4. Could the authors provide an analysis about training and inference efficiency, as the reflection stage would lead to much more overhead.

**Ethical Concerns:**

["NO or VERY MINOR ethics concerns only"]

**Final Justification:**

I acknowledge the strength of the proposed SRPO method and the clarity of the presentation. The concerns I raised have been adequately addressed, and I maintain my positive score.

**Limitations:**

They only verify their method on 7B and 32B Qwen models, without more exploration on other models with different frameworks and parameters.
They lack a systematic analysis on the training and inference efficiency, as their method will cause more overhead.

**Paper Formatting Concerns:**

In line 318, “figure” --> “Figure”.
In line 310, “Figure 3 The left part”-->“The left part of Figure 3”.

**Quality:**

3

**Strengths And Weaknesses:**

Strengths
a) Clarity: The paper is well-organized and clearly written. The framework is described in detail and the authors provide sufficient information about it.
b) Well-motivated: The paper's motivation is well-articulated, clearly identifying the limitations of local dependency in MLLM reasoning, often resulting in redundant, incoherent, or incorrect outputs
Weaknesses
a) Limited discussion of training efficiency: This paper utilized a two-stage training, but lack detailed discussion of the training cost of each stage.
b) Training data: Although the proposed method achieved good results compared with other components, it may result from the collection of training data also utilized by others. It will be beneficial to make a further ablation study about this.

---

> ### Author Rebuttal · Authors · 2025-07-30
>
> ### **Response to Reviewer J4Dj**
>
> We sincerely appreciate the reviewer’s positive and constructive feedback. Below, we systematically address the stated weaknesses and specific questions.
>
> > **Response to Weakness (a) and Question 4. Training & inference efficiency:**
>
> We thank the reviewer for the constructive feedback regarding training efficiency. Below, we provide the measured wall‑clock time for each SRPO training stage under both 7B and 32B settings using the OpenRLHF framework.
>
> #### **Table a. Training Efficiency Overview**
>
> | Training Stage               | GPUs         | Wall-Time  | Notes                        |
> |---------------------|--------------|------------|------------------------------|
> | Reflection-SFT (7B) | 8×H100       | 3.5 hours  | 10k samples, 1 epoch         |
> | Reflection-RL (7B)  | 8×H100       | 31.2 hours   | 37k samples, 500 steps       |
> | GRPO-RL (7B)  | 8×H100       | 25.8 hours   | 37k samples, 500 steps       |
> | Reflection-SFT (32B)| 4×8×H100     | 4.7 hours  | 10k samples, 1 epoch         |
> | Reflection-RL (32B) | 4×8×H100     | 45.1 hours   | 37k samples, 500 steps       |
>
> As shown in Table a, despite the two-stage design, **SRPO remains cost‑efficient**. The SFT stage completes in less than 5 hours even for 32B models, and the RL stage is comparable in duration to standard GRPO training. The additional rollout overhead introduced by the reflection segment is marginal, making SRPO’s overall compute footprint practical and scalable for real‑world deployment.
>
>
> #### **Table b. Inference Efficient Overview**
>
> | Inference Stage               | GPUs         | Wall-Time  | Notes                        |
> |---------------------|--------------|------------|------------------------------|
> | GRPO-RL (7B)  | 1×H100     | 45.6min  | 1K test samples, avg response=355.4 tokens   |
> | SRPO (7B)  | 1×H100     | 60.5min  | 1K test samples, avg response=502.3 tokens   |
>
> Table b shows inference efficiency, we evaluated reasoning latency on the MathVista test‑mini split (~1k examples), comparing **SRPO‑7B** with **Qwen2.5‑VL‑7B + GRPO** trained on the same RL data. Using a single H100‑80G GPU and the vLLM framework, SRPO requires only 1.33× more wall‑time and produces 1.41× more tokens on average. This modest overhead is controlled by our brevity‑aware reward $ f_{\text{len}} $, which promotes concise yet effective reflection. As shown in Table 3 of the paper, SRPO improves MathVista accuracy by **+3.5 points** and achieves an average **+3.7 gain** across benchmarks, showing that the slight inference cost is well justified by significant gains in reasoning quality.
>
> > **Response to Weakness (b) and Q4. Self-reflection Training data Ablation:**
>
> We thank the reviewer for raising this important point. To verify that SRPO’s performance gains stem from our reflection-aware training design rather than simply the training data, we conducted additional ablation experiments comparing Self-Reflection SFT and Plain-CoT SFT under identical data sources and teacher models (GPT-o4-mini).
>
> We constructed a Plain-CoT SFT baseline by removing the `<reflection>` segments from the same dataset and prompts. Additionally, the CoT reasoning paths were generated by GPT-4-mini given the question and the answer. We also applied GRPO to both the Plain-CoT and Reflection-SFT models. The results are summarized below:
>
> | Method                           | MathVista | MathVerse | MathVision | MMMU-Pro | Physics | Avg.  |
> |----------------------------------|-----------|-----------|-------------|----------|---------|--------|
> | Qwen‑2.5‑VL-7B                   | 68.2      | 46.3      | 25.1        | 36.9     | 45.4    | 44.4   |
> | Plain-CoT SFT                    | 69.1      | 47.2      | 26.4        | 36.2     | 47.6    | 45.3   |
> | Reflection-SFT (ours)           | 70.3      | 48.2      | 27.2        | 38.7     | 48.5    | 46.6   |
> | Plain-CoT SFT + GRPO            | 73.6      | 54.2      | 30.6        | 40.6     | 58.0    | 51.4   |
> | Reflection-SFT + Reflection-RL (full SRPO)  | **75.8**  | **55.8**  | **32.9**    | **42.3** | **60.6**| **53.5** |
>
> **Findings:**
> - **Plain-CoT SFT vs. Reflection-SFT**: Adding structured reflection yields an average gain of 1.3 points, indicating that performance improvements are not solely due to data coverage or quality.
>
> - **Plain-CoT SFT + GRPO vs. SRPO (full)**: When both variants are trained with RL, SRPO achieves an additional 2.1 gain, confirming that the reflection-aware reward design plays a critical role in enabling effective reasoning refinement.
>
> These results validate the importance of both our reflection-format data and the self-reflection RL stage in driving SRPO’s improvements.
>
> > **Response to Q1. Justification for the “first solution → reflection → second solution” structure:**
>
> We appreciate the reviewer’s thoughtful question. We clarify the rationale as follows:
>
> - **Concrete anchor for reflection**: Current MLLMs cannot introspect past hidden states once tokens are emitted. Producing an initial answer exposes an explicit artifact for the reflection to critique. Without this anchor, reflections would have to operate on inaccessible or implicit reasoning, making error localization impractical.
>
> - **Stable and decomposable reward signals**: Our reward design separates two components: (1) **R_accuracy**, which assesses the correctness of the *first answer* and provides immediate, low-variance feedback; and (2) **I_eff**, which evaluates whether the *second answer* improves upon the first, enabling direct supervision of reflection quality. If reflection were interleaved step-by-step, computing the reward would require aligning partial thoughts with the ground truth, leading to high credit-assignment variance and unstable RL.
>
> - **Avoiding reward hacking**: Training with an explicit “first → reflection → second” structure constrains the space of possible redundant  exploitation strategies, making it more robust than unconstrained step-wise reflection where the model may manipulate reasoning to game rewards.
>
> > **Response to Q2. Role of the Second Solution**:
>
> Thank you for the question. The second solution is not used solely for computing the reflection reward, it serves three critical purposes:
>
> -  **Final task output**: During validation and benchmark evaluation, only the second answer is scored. This ensures that the reflection must result in a self-consistent and correct final answer; otherwise, the model receives punished reward. (see equation (8) and Section 4.4 Reasoning Qualitative Analysis, lines 312–318).
>
> - **Gradient carrier for reflection learning**: SRPO applies token-level updates across the full sequence (answer 1 → reflection → answer 2). Although **R_accuracy**​ is tied to answer 1, tokens in answer 2 are still reinforced via: the shared task reward, and the **I_eff​** reward when answer 2 improves or preserves correctness. Thus, answer 2 is directly optimized to be both correct and concise.
>
> - **Separation of detection and repair**: Anchoring **R_accuracy**​ to answer 1 isolates “error detection”, while **I_eff​** on answer 2 captures “repair effectiveness”. This disentanglement would be lost if only answer 2 were rewarded, obscuring whether improvements came from better initial reasoning or reflection.
>
> > **Response to Q3. Determination of T_target**:
>
> Thank you for pointing this out. We define T_target​ as 2×  the length of the original first-think response, and T_max​ as 2.5× the original length. This sets a soft constraint for the combined reflection and revised reasoning to remain concise while allowing sufficient flexibility. We will clarify this definition explicitly in the main paper in the revised or arxiv version.
>
> > **Response to limitation:**
>
> We appreciate the reviewer’s constructive comment. As noted in our **Appendix Limitation section**, our current experiments focus on dense models (7B and 32B), consistent with prior works like RL-based reasoning in LLMs [1,2,3]. Extending SRPO to other architectures, such as large-scale MoE models or unified generation-understanding models, is a promising direction we plan to pursue in future work.
>
> [1] Cognitive Behaviors that Enable Self-Improving Reasoners, or, Four Habits of Highly Effective STaR
>
> [2] Does Reinforcement Learning Really Incentivize Reasoning Capacity in LLMs Beyond the Base Model?
>
> [3] Rethinking Reflection in Pre-Training
>
> > **Response to Paper Formatting Concerns**:
>
> We appreciate the reviewer’s kind reminder and will correct these minor typos in the revised version.

---

> > ### Author Response · Authors · 2025-08-04
> >
> > Dear Reviewer J4Dj, Your thoughtful review of our work is profoundly appreciated. We recognize the dedication it takes to provide such feedback. If there are areas in our response that need further clarification, or if additional questions arise, we stand ready to engage and offer any necessary assistance.

---

> > ### Comment · Reviewer_J4Dj · 2025-08-07
> >
> > I appreciate the authors’ detailed response. My concerns have been adequately addressed, and I maintain my positive score.

---

> > > ### Author Response · Authors · 2025-08-07
> > > **Thank you, Reviewer  J4Dj**
> > >
> > > Dear Reviewer  J4Dj,
> > >
> > > Thank you for your valuable feedback and kind support — your insights have been greatly in strengthening our work. We wish you continued success in your endeavors!
> > >
> > > Best regards, Team 12027

---

> ### Author Response · Authors · 2025-08-06
> **Any New Comments Would be Greatly Appreciated**
>
> Hi, Reviewer J4Dj!
>
> We sincerely appreciate your  thoughtful and encouraging review, and acknowledged our paper's clear motivation, detailed method description, and strong empirical results. We thank you for recognizing the novelty and clarity of our framework.
>
> We have carefully addressed all raised concerns in our rebuttal. Specifically, we have included detailed training cost breakdowns and efficiency analyses for both 7B and 32B models, as well as clarified the necessity of the “first solution → reflection → second solution” process. Additional ablations have also been added to isolate the effect of reflection structure and reward components.
>
> We truly value your thoughtful review and hope to gain your continued support. If there are any parts of our submission that would benefit from additional clarification or elaboration to assist your evaluation, please feel free to let us know, we would be glad to provide a further detailed response.
>
> Thank you again for your thoughtful review and support.
>
> Best wishes and regards,
>
> Team 12027

---

### Official Review · Reviewer_mGVt · 2025-07-01

**Clarity:** 3
**Significance:** 3
**Originality:** 3
**Rating:** 4
**Confidence:** 3

**Summary:**

This paper proposes Multimodal Self-Reflection Enhanced Reasoning with Group Relative Policy Optimization (SRPO), a two-stage reflection-aware RL framework specifically designed to improve reasoning in MLLMs. The proposed model demonstrates strong performance on multimodal reasoning benchmarks.

**Questions:**

Could the authors provide more examples from the constructed dataset?

Regarding the reproduced results in Table 1, were these results originally reported in the referenced paper? If so, could the corresponding values be included in the table for comparison?

**Ethical Concerns:**

["NO or VERY MINOR ethics concerns only"]

**Final Justification:**

The author's rebuttal has addressed most of my concerns. I keep my original rating as 4.

**Limitations:**

yes

**Quality:**

3

**Strengths And Weaknesses:**

Strengths:
1. The motivation of the paper is clearly articulated, and the proposed approach is simple yet effective.
2. Extensive ablation studies demonstrate the effectiveness and generalizability of the proposed method.
3. Experiments validated the capabilities of both the 7B and 32B models, with the 32B model achieving comparable performance to some closed-source models on certain benchmarks.

Weaknesses:
1. The presentation of the paper needs improvement, as the spacing of many figures and tables requires adjustment, which affects the overall readability of the paper.
2. In my view, the construction of the dataset is equally important; however, the presentation of the training data is relatively limited in the paper.

---

> ### Author Rebuttal · Authors · 2025-07-30
>
> ### **Response to reviewer mGVt**
>
> We thank Reviewer mGVt for the positive assessment and valuable suggestions.
>
> > **Response to W1. Formatting:**
>
> We acknowledge the spacing issues that arose from automatically scaling wide figures. In the new revised version we will move Fig. 2 and Tables 1–2 to the top of the next page and re‑balance texts and figure/tables space.
>
> > **Response to W2 and  Q1. Training‑data transparency:**
>
> Thank you for the constructive suggestion. Due to space constraints, we briefly described the training data in the Methods section. A detailed explanation is provided in **Appendix B.1**, where we outline the construction of both the Self-Reflection SFT and RL datasets, including the specific open-source sources used and the prompting templates detailed in **Appendix B.3**.
>
> Specifically, for the Self-Reflection SFT dataset, we follow the prompt format defined in *Appendix B.3: Prompt Template for Self-Reflection Generation*, and structure the data in the form of *first solution → reflection → refined solution*. For the RL dataset, we unify several open-source datasets into a consistent format containing *image, question, and answer label*.
>
> As NeurIPS rebuttal rules prohibit modifying the PDF or adding visual content at this stage, we commit to releasing all training datasets, along with the full source code for data selection and construction pipelines for both SFT and RL, in the new-revised version.
>
> > **Response to Q2. Clarifying Table 1 sources:**
>
> Thank you for the helpful question. For the results reported in Tables 1 and 2, baseline numbers are taken from the official reports whenever available. For datasets not covered in those reports, we reproduce results using the authors' released checkpoints and hyperparameters under our unified evaluation setup. These reproduced results are marked with a † symbol for clarity.
>
> We will include this clarification in the revised version. We appreciate the reviewer’s suggestion.

---

> > ### Comment · Reviewer_mGVt · 2025-08-02
> >
> > Thanks for the author's rebuttal. I maintain my positive rating.

---

> > > ### Author Response · Authors · 2025-08-02
> > > **Thank you, Reviewer mGVt**
> > >
> > > Dear Reviewer mGVt,
> > >
> > > Thank you for your thoughtful feedback and support. Your insights have greatly improved our work. Wishing you continued success!
> > >
> > > Best regards,
> > > Team 12027

---

### Official Review · Reviewer_PKxa · 2025-07-01

**Clarity:** 3
**Significance:** 2
**Originality:** 2
**Rating:** 4
**Confidence:** 4

**Summary:**

The paper presents SRPO, a two-step reinforcement-learning framework that aims to sharpen the reasoning skills of multimodal large language models by baking in deliberate self-reflection.

In the first step—supervised fine-tuning—a strong teacher model critiques the base model’s reasoning paths and leaves “reflective breadcrumbs” in a custom dataset. The second step rolls out GRPO, pairing it with a reflection-aware reward: models earn points not only for the right answer, but also for producing short, sharp self-critiques that actually improve their second-try reasoning.

Across benchmarks such as MathVista and MMMU-Pro, SRPO consistently tops other open-source MLLMs and even holds its own against several well-guarded proprietary models.

Overall, performing SFT first using data with self-correction or reflection does make RL training easier—a widely adopted practice that aligns with expectations. Therefore, it’s hard to consider this aspect as an original contribution of the paper.

**Questions:**

Please see the weaknesses.

**Ethical Concerns:**

["NO or VERY MINOR ethics concerns only"]

**Final Justification:**

I adjusted my score to borderline accept.

**Limitations:**

Please see the weaknesses.

**Quality:**

2

**Strengths And Weaknesses:**

## Strengths

**1.Novel Reward Mechanism**

The core strength of this work lies in the design of its reflection-aware reward function. The effectiveness indicator, Ieff, which explicitly rewards or penalizes a reflection based on its functional impact on the final answer, is a clever and direct approach to promoting meaningful self-correction and mitigating the risk of reward hacking through verbose or vacuous reflections.

**2.Strong Empirical Results**

The paper presents compelling and comprehensive experimental results across a wide range of challenging multimodal reasoning benchmarks (Tables 1 and 2). SRPO consistently achieves state-of-the-art performance among open-source models and shows a significant leap over the base Qwen-2.5-VL models, validating the effectiveness of the proposed method.

## Weaknesses

**1.Inflated claims about novelty**

The paper pitches the two-stage “SFT → RL” pipeline as a fresh framework, yet that recipe is already standard fare for steering both uni- and multimodal LLMs (see, e.g., [9, 15, 36, 577]). GRPO is likewise lifted straight from prior work [4]. What’s genuinely new here is the way the authors build a reflection-centric SFT set and, more importantly, how they cook up a reward that nudges the model to critique—and then improve—itself. Those pieces are worthwhile, but the manuscript oversells the conceptual leap. This feels more like a smart piece of reward engineering atop a familiar scaffold than a brand-new paradigm.  Consequently, it is unclear why SRPO should be considered a research advance rather than an engineering variant of prior art.

**2.SFT ablation muddied by teacher-model distillation**

The “cold-start” phase lets GPT-4o-mini write the reflections, effectively distilling a stronger model into the student. Table 3 (“w/o Self-Reflection SFT”) confirms this stage matters, but the experiment doesn’t tease apart *why*. Is the gain coming from the fancy < reflection > wrapper, or simply from gulping down better reasoning traces? We already know that fine-tuning on a superior teacher’s outputs can lift a model’s scores, reflection or no reflection. A cleaner test would build a control SFT set of equal size, generated by the same teacher, but containing plain CoT answers—no explicit reflection blocks. Training SRPO on that control and comparing it to the proposed version would reveal whether the reflective format itself drives the lift, or if standard distillation explains most of the bump. Without this contrast, the headline improvement could just be old-fashioned teacher-student transfer masquerading as something novel.

**3.Insufficient Ablation**

3.1 The main technical contribution in the RL stage is the composite reward function, Equation 6. However, the ablation provided ("w/o Self-Reflection RL") removes this entire composite reward at once, rather than dissecting its components. This makes it impossible to attribute the gains to specific design choices. A more rigorous analysis, crucial for a methods paper, would involve ablating each component individually. Furthermore, the choice of the weighting hyperparameter α=0.1 is presented without justification; a sensitivity analysis on this parameter is needed to assess the robustness of the reward design.

3.2 All component analyses—like the effect of reflection-based SFT and RL—are done solely on the 7B model. Yet the strongest results come from the 32B variant, where no such breakdown is provided. This is a key gap: scaling often changes what matters most, and insights from 7B don’t always transfer.

**4.Reflection quality not evaluated**

The core claim is that SRPO yields “concise and cognitively meaningful” reflections. However, evaluation remains wholly *extrinsic*—answer accuracy. No human study, automatic reflection-quality metric, or analysis of redundancy versus utility is provided. Figure 3 gives two cherry-picked examples, insufficient to establish systematic improvement and leaving open the possibility of reward hacking through superficial length tuning.

---

> ### Author Rebuttal · Authors · 2025-07-30
>
> ### **Response to reviewer PKxa**
>
> We thank Reviewer PKxa for the positive feedback on our reward mechanism and empirical results.
>
> > **Response to Weakness 1. (Clarifying SRPO’s Contribution):**
>
> We thank Reviewer PKxa for the thoughtful comments. While the SFT → RL pipeline is widely used, **SRPO introduces key innovations that extend beyond this standard scaffold**:
>
> - **Explicit and Measurable Reflection in Both Stages**: SRPO is, to our knowledge, the first to:**(i)** incorporate explicit reflection structure during SFT, teaching the multimodal LLM to critique its own reasoning; and
> **(ii)** reinforce reflection quality during RL using the reflection‑aware indicator **I_eff**, which directly links reflection effectiveness to answer improvement.   Unlike prior work that either fine‑tunes only on CoT data or applies RL without explicitly revisiting and correcting prior reasoning, SRPO makes reflection both **structurally grounded** and **quantitatively rewarded**, elevating it to a core trainable mechanism for improving reasoning accuracy.
> - **Reflective Structure vs. Two‑Step Thinking**: To evaluate the effectiveness of the proposed *think–reflect–rethink* pattern, we compared SRPO to a GRPO‑based two‑step thinking baseline that generates two consecutive `<think>...</think>` steps without middle reflection. As shown below, SRPO without Self‑Reflection SFT already outperforms the two‑step baseline, while the full SRPO achieves the best performance, underscoring the importance of combining Self‑Reflection SFT with reflection‑aware RL.
>
> #### **Table a: Comparison between SRPO and GRPO with Two‑Step Thinking**
>
> | Methods                      | MathVista | MathVerse | MathVision | MMMU-Pro | Physics | Avg.  |
> |------------------------------|-----------|-----------|-------------|----------|---------|-------|
> | *Ours – SRPO‑7B*             | **75.8**  | **55.8**  | **32.9**    | **42.3** | **60.6**| **53.5** |
> | SRPO w/o Self‑Reflection SFT | 74.2      | 53.3      | 30.3        | 39.7     | 58.6    | 51.2   |
> | GRPO + Two‑Step Thinking     | 73.5      | 53.6      | 30.6        | 40.3     | 53.6    | 50.3   |
>
> - **Contribution of the Reflection‑Aware Reward I_eff**
>
> To isolate the impact of I_eff , we removed it while keeping all other hyperparameters unchanged. This resulted in a consistent drop across benchmarks, confirming that the reflection‑aware reward contributes substantially to SRPO’s improvements beyond simple brevity shaping.
>
> #### **Table b: Contribution of I_eff**
>
> | Methods                                  | MathVista | MathVerse | MathVision | MMMU-Pro | Physics | Avg.  |
> |------------------------------------------|-----------|-----------|-------------|----------|---------|-------|
> | *Ours – SRPO‑7B*                         | **75.8**  | **55.8**  | **32.9**    | **42.3** | **60.6**| **53.5** |
> | w/o Reflection‑Aware Reward I_eff | 74.2      | 53.3      | 30.3        | 39.7     | 58.6    | 51.2   |
>
> > **Response to W2: Is the SFT improvement solely due to teacher distillation?**
>
> To disentangle the impact of our reflection format from generic teacher–student transfer, we conducted the following control experiments:
>
> #### **Table c: Ablation Study of Self‑Reflection SFT on the 7B model**
>
> | Methods                                    | MathVista | MathVerse | MathVision | MMMU-Pro | Physics | Avg.  |
> |--------------------------------------------|-----------|-----------|-------------|----------|---------|-------|
> | Qwen‑2.5‑VL‑7B                             | 68.2      | 46.3      | 25.1        | 36.9     | 45.4    | 44.4  |
> | Plain‑CoT SFT                              | 69.1      | 47.2      | 26.4        | 36.2     | 47.6    | 45.3  |
> | Reflection‑SFT (ours)                      | 70.3      | 48.2      | 27.2        | 38.7     | 48.5    | 46.6  |
> | Plain‑CoT SFT + GRPO                       | 73.6      | 54.2      | 30.6        | 40.6     | 58.0    | 51.4  |
> | Reflection‑SFT + Reflection‑RL (full SRPO) | **75.8**  | **55.8**  | **32.9**    | **42.3** | **60.6**| **53.5** |
>
> **Setup**: All models use the same teacher (GPT‑o4‑mini), dataset size (10k), optimizer, and number of training epochs. The only differences are the inclusion of `<reflection>` segments and whether reflection‑aware RL is applied.
>
> **Findings:**
>
> - **Plain‑CoT SFT vs. Reflection‑SFT**: Adding explicit reflection segments yields an average +1.3 gain, showing that structured reflection, not just teacher distillation, contributes meaningfully to performance.
>
> - **Reflection‑SFT + GRPO vs. SRPO (full)**: When both undergo RL, SRPO achieves an additional +2.1 gain by aligning rewards with reflection effectiveness, underscoring the importance of our reflection‑aware reward design.
>
> > **Response to Weakness 3.1. (Composite‑reward ablation and α‑sensitivity):**
>
> Below we (i) disentangle each reward component and (ii) study the robustness of the mixing coefficient α.
> - **(i) Ablation of  Reflection-Aware RL Components.**
>
> | Methods                  | MathVista | MathVerse | MathVision | MMMU-Pro | Physics | Avg.  |
> |--------------------------|-----------|------------|-------------|----------|---------|--------|
> | full SRPO    | **75.8**  | **55.8**   | **32.9**    | **42.3** | **60.6**| **53.5** |
> | w/o Self-Reflection RL     |70.3   | 48.2    | 27.2     | 38.7  | 48.5  | 46.6 |
> | -no Length Reward (f_len(·))     |75.3  | 56.2    | 32.4     | 41.7 | 60.1  |  53.1 |
> | -no Reflection Reward (I_eff)    |73.9   | 54.7   | 31.6     | 40.9  | 58.8  |  52.0 |
>
> **Findings:**  Removing **I_eff** drops the 5‑task average by 1.5 points, showing it is the key driver of SRPO’s gains. Removing **f_len(·)** has a smaller effect (–0.4). Dropping the entire Self‑Reflection RL stage causes the largest drop (–6.9), underscoring the importance of SRPO’s integrated design.
>
> - **(ii) Sensitivity to the reflection length reward coefficient α**
>
> Due to rebuttal time and compute limits, we additionally trained with α=0.05 and α=0.3 and tested the dataset from three domains. Results show α=0.05 performs similarly to 0.1, while α=0.3 degrades accuracy, indicating that overemphasizing length reward weakens the reflection signal. Therefore, values near 0.1 offer a balanced trade-off, suggesting that length regularization primarily helps control response length and stabilize training, with limited impact on accuracy.
> | Ratio                 | MathVerse | MMMU-Pro | Physics | Avg.  |
> |--------------------------|-----------|------------|-------------|------------|
> |         α=0.05                     |       55.7         |         41.9           |        61. 1   |  52.9     |
> |         α=0.1(Ours)             |      55.8      |       42.3       |       60.6       |       53.5    |
> |         α=0.3                       |       55.1      |     42.0         |        59.7       |       52.2     |
>
> > **Response to W3.2 (Ablation study of 32B models):**
>
> We evaluate SRPO on 32B with the same ablation setup as 7B. SRPO‑32B outperforms Qwen‑32B‑VL + GRPO, confirming the value of explicit reflection in SFT and RL. Dropping either component clearly hurts performance, consistent with the 7B trends.
> | Method                            | MathVerse | MMMU-Pro | Physics | Avg.   |
> |----------------------------------|-----------|----------|---------|--------|
> | Qwen-2.5-VL-32B + GRPO           | 56.0      | 49.0     | 57.0    | 54.0   |
> | **SRPO-32B (ours)**              | 58.9      | 51.3     | 64.2    | 58.1  |
> | w/o Self-Reflection SFT     | 56.3      | 48.2     | 62.1    | 55.5  |
> | w/o Self-Reflection RL       | 50.9      | 47.0     | 51.4    | 49.8  |
> | – no Length Reward (f_len(·))    | 59.3      | 50.7     | 63.7    | 57.9  |
> | – no Effectiveness Reward (I_eff)| 57.7      | 50.0     | 62.5    | 56.7  |
>
> > **Response to Question 1 and weakness 5. (Qualitative & quantitative reflection analysis)**:
>
> Due to space limitations in the main text, Figure 3 shows abbreviated samples. We provide full reflection traces from both RL training and downstream evaluation in Appendix B.4, which demonstrate the model’s ability to self-reflect and revise its reasoning steps. Following your suggestion, we further conducted a human evaluation of reflection quality:
>
> **1. Human Expert Study for Reflection Quality:**
> We randomly selected 100 MathVista test samples (answered by SRPO-7B). Two additional senior PhD students (NLP/LLMs) rated each reflection on a 0–3 scale:
> • 3 = Highly effective • 2 = Partially effective • 1 = Redundant • 0 = Detrimental
> We also measured the Wrong Answer Fix Rate: how often initially incorrect answers were corrected after reflection.
> #### Table d. Human Expert Evaluation
> | Metric                          | Human Expert 1 | Human Expert 2 |
> |--------------------------------|----------------|----------------|
> | Effective Reflection Rate (score ≥ 2) | 73%           | 69%           |
> | Redundancy Rate (score = 1)           | 9%            | 11%           |
> | Detrimental Rate (score = 0)          | 3%            | 1%            |
> | Wrong Answer Fix Rate                 | 39%           | 39%           |
>
> This shows that 70% of reflections are helpful. Out of 100 questions, the initial solution was incorrect for 33 cases. Among them, 13 were corrected after reflection and a second attempt, resulting in a wrong answer fix rate of 39%.
>
> **2. LLM-as-a-Judge Evaluation (GPT‑4o) for Reflection Quality:**
>
> Each reflection was scored on 4 dimensions (0–5 scale): logical flaws, missing assumptions, clarity, and actionable suggestions following **B.3 Prompt Template. Prompt Template for Self-Reflection Generation**.
>
> #### **Table e. LLM-as-a-Judge Evaluation (GPT‑4o)**
> | Model     | Logic | Missing | Clarity | Suggestions | Avg Quality |
> |-----------|--------|----------|---------|--------------|----------|
> | SRPO-7B   | 4.1    | 3.9      | 4.0     | 3.6          | **3.9** |
>
> These results confirm high-quality reflections from both human and LLM perspectives.

---

> > ### Comment · Reviewer_PKxa · 2025-08-01
> >
> > Thank you for the detailed response, which has addressed the majority of my concerns. I will re-evaluate the submission accordingly and will consider adjusting my rating.

---

> > > ### Author Response · Authors · 2025-08-01
> > > **Many thanks, Reviewer PKxa**
> > >
> > > Dear Reviewer PKxa,
> > >
> > > Thank you for your valuable comments and contributions to this work. We will incorporate your suggestions and experiments into the main text and the appendix. Your feedback and support have greatly improved our paper. We sincerely appreciate your efforts and wish you success in your future research.
> > >
> > > Bests,
> > >
> > > Your friends,
> > >
> > > Team 12027

---

> > > ### Author Response · Authors · 2025-08-05
> > >
> > > Dear Reviewer PKxa, Many thanks for your guidance and postive comment after again. Before the discussion window closes, we’re happy to supply any additional clarification or small-scale checks you find useful.

---

> > > ### Author Response · Authors · 2025-08-06
> > >
> > > Hi, Reviewer PKxa!
> > >
> > > We sincerely thank you for taking the time to provide extremely detailed reviews and highly insightful suggestions. In particular, your recommendations **(1)** regarding disentangling the effect of reflection format vs. teacher distillation, **(2)** granular ablations of the reflection-aware reward, **(3)** full component analysis for the 32B setting, **(4)** qualitative and quantitative analyses of reflection quality have all significantly enhanced the quality and completeness of our research.
> > >
> > > We appreciate that you have completed the Mandatory Acknowledgement. As a highly important and respected reviewer, we sincerely hope to receive your favorable consideration and further support. If there’s anything you would like us to further explain or supplement in order to gaining your support, please kindly let us know—we’re ready to respond immediately.
> > >
> > > Thank you once again for your time, dedication and support!
> > >
> > > Best wishes and regards,
> > >
> > > All authors of Submission 12027

---

### Author Response · Authors · 2025-08-09
**Heartfelt Thanks to All Reviewers and the AC Team**

Dear AC, SPC, PC, and Reviewers,

As the author–reviewer discussion period concludes, we would like to once again express our sincere gratitude to all reviewers, the AC, SPC, and PC for your time, dedication, and invaluable contributions. Your efforts not only helped improve our work but also strengthened the NeurIPS community as a whole.

We are especially grateful to Reviewers `PKxa`, `mGVt`, `J4Dj`, and `wAHj` for their constructive, insightful and feedback, including:

- Disentangling the effect of reflection format vs. teacher distillation (thanks to reviewers `PKxa` and `J4Dj`).
- Granular ablations of the reflection-aware reward (thanks to reviewer `PKxa` ).
- Full component analysis for the 32B setting (thanks to reviewer `PKxa`, `J4Dj`, and `wAHj` ).
- Qualitative and quantitative analyses of reflection quality (thanks to reviewers  `PKxa`, `J4Dj`, and `wAHj` ).
- Efficiency analyses for both 7B and 32B models (thanks to reviewers  `J4Dj` and `wAHj` ).
- Training data analysis  (thanks to reviewer `mGVt` ).
- Thoughtful suggestions on potential future directions  (thanks to reviewer `wAHj` ).

These comments have been instrumental in refining our work. We highly value these insightful recommendations and have addressed them thoroughly during the discussion, and we will incorporate the corresponding improvements into the revised version. We sincerely appreciate the positive feedback provided by all reviewers.

Finally, we look forward to having a chance to meet everyone in person at NeurIPS in San Diego, to continue the conversation on advancing RL for multimodal LLM reasoning and driving progress in this exciting research direction together.

Once again, we deeply appreciate the time and effort you have dedicated to improving and supporting this paper and to contributing to the advancement of the NeurIPS community.

With sincere thanks and best regards,
All authors of Submission 12027

---

### Decision · Program_Chairs · 2025-09-17

**Decision:**

Accept (poster)

**Comment:**

This paper proposes SRPO, which formulates “reflection” as a learnable skill for multimodal reasoning through a two-stage pipeline: (1) a reflection-oriented SFT stage that seeds models with initial-solution → reflection → gold-solution examples, and (2) GRPO with reflection-aware rewards that enforce structured outputs, reward first-answer accuracy, explicitly score the reflection’s usefulness, and softly encourage brevity.

The method is well-motivated, and experiments on Qwen-2.5-VL 7B/32B show consistent improvements over strong GRPO baselines across MathVista, MathVerse, MathVision, MMMU(-Pro), EMMA, and MMK12. Overall, all of us lean toward acceptance.

The remaining unsolved concerns:
- The generalizability of the method is not fully established, as the evaluation is largely limited to STEM-focused tasks
- One reviewer questioned the level of scientific novelty, suggesting that the work leans more toward engineering practice than fundamental contribution
- The training data plays a crucial role in achieving strong results. The ablation study only partially addresses this issue and does not provide sufficient evidence regarding the method’s robustness to data variations.